# Androgen receptor and MYC equilibration centralizes on developmental super-enhancer

Haiyang Guo[1,2,3,15], Yiming Wu[4,5,15], Mannan Nouri [4], Sandor Spisak [6,7], Joshua W. Russo [4], Adam G. Sowalsky [8], Mark M. Pomerantz[6], Zhao Wei[9], Keegan Korthauer[10], Ji-Heui Seo[6], Liyang Wang[4], Seiji Arai [4,11], Matthew L. Freedman[6,7,12], Housheng Hansen He [13,14✉], Shaoyong Chen [4✉] & Steven P. Balk [4✉]

Androgen receptor (AR) in prostate cancer (PCa) can drive transcriptional repression of multiple genes including *MYC*, and supraphysiological androgen is effective in some patients. Here, we show that this repression is independent of AR chromatin binding and driven by coactivator redistribution, and through chromatin conformation capture methods show disruption of the interaction between the *MYC* super-enhancer within the *PCAT1* gene and the *MYC* promoter. Conversely, androgen deprivation in vitro and in vivo increases MYC expression. In parallel, global AR activity is suppressed by MYC overexpression, consistent with coactivator redistribution. These suppressive effects of AR and MYC are mitigated at shared AR/MYC binding sites, which also have markedly higher levels of H3K27 acetylation, indicating enrichment for functional enhancers. These findings demonstrate an intricate balance between AR and MYC, and indicate that increased MYC in response to androgen deprivation contributes to castration-resistant PCa, while decreased MYC may contribute to responses to supraphysiological androgen therapy.

[1] Department of Clinical Laboratory, The Second Hospital, Cheeloo College of Medicine, Shandong University, Jinan 250033, Shandong, China. [2] Shandong Engineering & Technology Research Center for Tumor Marker Detection, Jinan 250033, Shandong, China. [3] Shandong Provincial Clinical Medicine Research Center for Clinical Laboratory, Jinan 250033, Shandong, China. [4] Hematology-Oncology Division, Department of Medicine, Beth Israel Deaconess Medical Center and Harvard Medical School, Boston, MA 02215, USA. [5] Institutes of Biomedical Sciences, Fudan University, Shanghai 200433, China. [6] Department of Medical Oncology, Dana Farber Cancer Institute, Harvard Medical School, Boston, MA, USA. [7] Center for Functional Cancer Epigenetics, Dana-Farber Cancer Institute, Boston, MA 02215, USA. [8] Laboratory of Genitourinary Cancer Pathogenesis, National Cancer Institute, National Institutes of Health, Bethesda, MD 20892, USA. [9] Department of Clinical Laboratory, Qilu Hospital of Shandong University, Jinan 250012 Shandong, China. [10] Department of Statistics, University of British Columbia, Vancouver, BC, Canada. [11] Department of Urology, Gunma University Hospital, Maebashi, Gunma, Japan. [12] The Eli and Edythe L. Broad Institute, Cambridge, MA 02142, USA. [13] Department of Medical Biophysics, University of Toronto, Toronto, ON, Canada. [14] Princess Margaret Cancer Center, University Health Network, Toronto, ON, Canada. [15]These authors contributed equally: Haiyang Guo, Yiming Wu. ✉email: hansenhe@uhnresearch.ca; schen@bidmc.harvard.edu; sbalk@bidmc.harvard.edu

The androgen receptor (AR) plays a central role in prostate cancer (PCa) development, and androgen deprivation therapy (ADT, medical or surgical castration) to suppress AR activity is the standard treatment for metastatic PCa, but tumors invariably recur (castration-resistant prostate cancer, CRPC). Many respond to agents that further suppress androgen synthesis such as abiraterone, or to direct AR antagonists (enzalutamide or apalutamide), but most men still relapse within 1–2 years. A subset of these relapsed tumors appear to be AR-independent, but the majority have persistently high levels of AR expression and activity. One mechanism driving this increased AR expression is amplification of the *AR* gene and an upstream *AR* enhancer, which occurs in the majority of cases[1–3]. Significantly, this upstream *AR* enhancer is activated primarily in CRPC[2]. A second frequent genomic alteration in CRPC is amplification of the *MYC* gene. Interestingly, the enhancer driving *MYC* expression in CRPC is also developmentally regulated and prostate-specific, and is strongly activated in CRPC[4,5]. The precise basis for the activation of these enhancers is not clear, but may be related to alterations that occur in the AR cistrome and transcriptome with PCa development and progression[6–8], and to a broad reactivation of developmental epigenomic programs found with progression to CRPC[9].

Interactions between AR and MYC proteins may also contribute to PCa development and progression. AR stimulation has been reported to suppress the expression of MYC in normal prostate epithelium, which may be a physiological mechanism through which AR drives terminal differentiation[10]. In contrast, AR has been found to increase MYC expression in several PCa cell lines, and in AR-positive apocrine breast cancer cells[10–12]. However, most studies have found that AR stimulation in PCa cells decreases expression of MYC, and this appears to be a direct effect on MYC transcription[13–15]. The molecular basis for this AR repression of MYC, and more broadly for AR-mediated repression of multiple additional genes, is less clear and may be through diverse mechanisms[11,16,17].

MYC overexpression can override androgen-mediated cell differentiation in normal prostate cells and drive androgen-independent proliferation in PCa cells[10,18]. The extent to which this is related to restoration of AR function is unclear as MYC overexpression has been reported to broadly suppress AR activity[19]. However, further data suggest that increased MYC in CRPC may alter the AR cistrome, as MYC overexpression causes some gain in AR binding sites, and the MYC motif is enriched in a subset of AR binding sites that are acquired in CRPC[7,19]. Finally, MYC has been reported to modulate AR splicing[20].

In this report, we show that increased AR expression markedly enhances androgen-mediated transcriptional repression, and that this repression is largely independent of AR binding to chromatin and is associated with coactivator redistribution. MYC is rapidly and dramatically downregulated by androgen in cells expressing high AR levels, and a subset of androgen-repressed genes are MYC regulated and suppressed due to reduced MYC. Conversely, suppression of androgen activity in vitro and in vivo increases MYC expression. We further find that MYC binding is also associated with a subset of androgen-stimulated genes, and that a substantial fraction of MYC binding sites overlap AR binding sites. Notably, these AR/MYC binding sites have markedly higher levels of H3K27 acetylation (H3K27ac) relative to AR alone binding sites, indicating that the AR/MYC sites are enriched for functional enhancers. Global AR activity is also enhanced or repressed by MYC downregulation or overexpression, respectively, also consistent with coactivator redistribution. Finally, we establish that androgen treatment represses MYC expression by disrupting the interaction between the prostate-specific *MYC* super-enhancer within the *PCAT1* gene and the *MYC* promoter.

Together these findings demonstrate an intricate link between AR and MYC function in PCa, wherein they both compete for coactivators and function cooperatively to maintain stable expression of genes regulating multiple cellular functions. Clinically, these findings indicate that increased MYC in response to androgen deprivation contributes to the development of CRPC, while decreased MYC may contribute to tumor regression in response to supraphysiological androgen therapy in men with CRPC.

## Results

**Global transcriptome assessment indicates an indirect AR transcriptional repression function.** Previous studies have shown that AR, in addition to its well-established function as a transcriptional activator, can also mediate transcriptional repression in cells expressing high levels of AR as occurs in CRPC. To assess effects of AR levels on the AR transcriptome we compared the LNCaP PCa cell line (LN, expressing moderate AR levels), the VCaP cell line (VC, which has an amplified *AR* gene and higher levels of AR expression), and LNCaP cells stably overexpressing exogenous AR (LA) (Fig. 1a, b). RNA-seq and microarray datasets comparing androgen-starved cells (cultured in medium with FBS that is charcoal dextran stripped to deplete steroids, CDS medium) versus DHT-stimulated cells both showed significantly more DHT-stimulated and DHT-repressed genes in VCaP versus LNCaP cells, and also an increase in the ratio of repressed to activated genes (Fig. 1c). DHT-stimulated and repressed genes in the AR-overexpressing LNCaP cell line were similarly greater than in parental LNCaP, indicating these differences were related to AR abundance (Fig. 1c).

Examination of overlapping DHT-stimulated versus DHT-repressed genes in the microarray and RNA-seq data also showed that the number of DHT-repressed genes was increased in the VCaP and AR-overexpressing LNCaP cells (Fig. S1a). Similarly, comparison of overlapping DHT-stimulated genes and DHT-repressed genes between the cells shows greater effects in the VCaP and LNCaP-AR cells (Fig. S1b). Consistent with the greater effects of DHT in the VCaP cells, AR ChIP-seq data show substantially more AR peaks in VCaP versus LNCaP cells (Fig. S1c). Notably, these AR peaks in LNCaP and VCaP cells had similar overall chromosomal distributions (Fig. S1d), and there was substantial overlap between the peaks (Fig. S1e), consistent with increased binding being driven by higher AR level in VCaP cells.

Notably, the majority of DHT-mediated transcriptional repression occurred at later times (10 and 24 h), suggesting indirect mechanisms, although this likely also reflects transcript stability for some genes (Fig. 1d). Transcriptional stimulation was similarly greater at the latter times, which we showed previously was due at least in part to a requirement for new protein synthesis to maximally drive AR transactivation function[21]. To further assess for direct effects we used binding and expression target analysis (BETA) to integrate the DHT-driven gene expression changes with genome-wide AR binding data, which was assessed by ChIP-seq in VCaP cells after 12 h of DHT stimulation[22]. This showed that AR binding (as assessed by ChIP-seq at 12 h after DHT-stimulation) was highly correlated with DHT-stimulated genes, but not with DHT-repressed genes (Fig. 1e). AR binding at repressed genes became significant only at the 24 h time point, indicating an indirect effect for the majority of DHT-repressed genes.

We also analyzed the overlap between AR and H3K27ac ChIP-seq data generated in VCaP cells cultured under basal conditions (10% FBS medium that was not androgen depleted)[22,23]. Consistent with the BETA showing an indirect effect of AR,

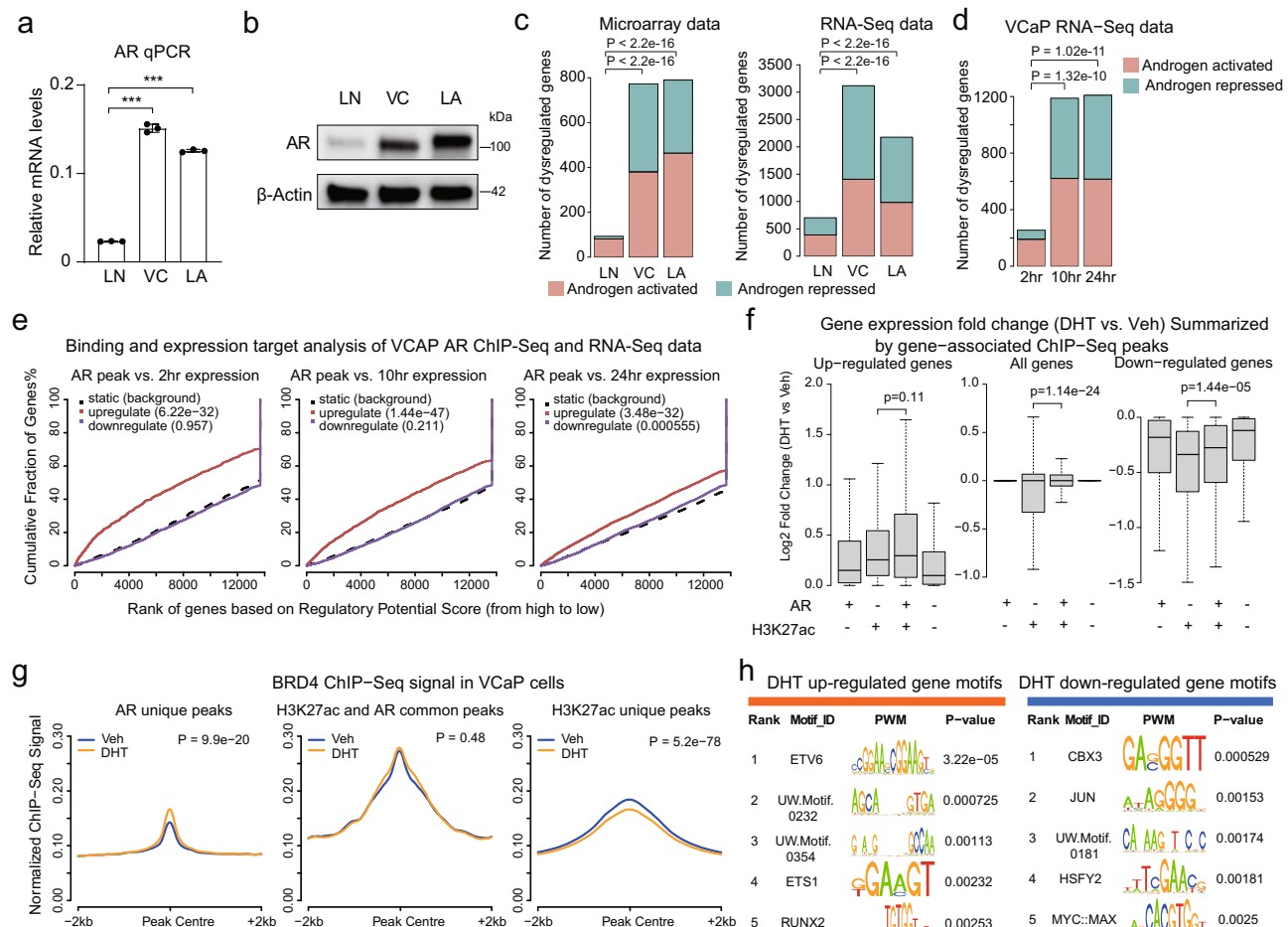

**Fig. 1 Global transcriptome assessment identifies a disassociation between AR and androgen-elicited transcriptional repression. a** RT-qPCR for AR mRNA in LNCaP (LN), VCaP (VC), and AR-overexpressing LNCaP (LNCaP-AR, or LA) cells. Data are mean values ± SD for three biologically independent samples. *P* values are two-sided Student's *t* test. ***, *P* < 0.001. **b** Western blot of AR in LNCaP, VCaP, and AR-overexpressing LNCaP cells, representative of three replicates. **c** Statistics of androgen-regulated genes. For microarrays, LNCaP is from Xu et al.[21] LNCaP-AR is from GSE62474. VCaP is from GSE62473. Cutoff for androgen-activated genes is fold change > 1.5 and adjusted *P* < 0.05. Cutoff for androgen repressed genes is fold change < 2/3 and adjusted *P* < 0.05. RNA-seq data for LNCaP is from Liang et al.[59]. LNCaP-AR and VCaP RNA-seq data were generated in this report. Cutoff for DHT-stimulated genes, fold change > 2 and adjusted *P* < 0.05; cutoff for DHT-repressed genes, fold change < 1/2 and adjusted *P* < 0.05. Chi-squared test determined whether the numbers of regulated genes are different. **d** Statistics of DHT-regulated genes at different time points in VCaP. Cutoff for androgen activated genes, fold change > 2 and adjusted *P* < 0.05; cutoff for DHT-repressed genes, fold change < 1/2 and adjusted *P* < 0.05. Chi-squared test determined if the ratio of regulated genes is different. **e** Binding and expression target analysis (BETA) for association between AR binding in VCaP cells assessed by ChIP-seq at 12 h after DHT-stimulation (GSE55062) and DHT-driven gene expression at different time points in VCaP cells. Red line indicates DHT-upregulated genes, purple line indicates DHT-downregulated genes. **f** Boxplots show expression fold changes, which were classified into AR only, H3K27ac only, H3K27ac/AR common, or None groups based on H3K27ac or/and AR occupancy at ±10 Kb regions of gene TSSs. Box limits, 1st and 3rd quartiles; whiskers, 1.5× inter-quartile range; center line, median. From the left to right, *n* = 1090, 7989, 591, 2892, 5742, 25829, 1876, 19010, 1527, 12663, 600, and 4351, respectively. *P* values by two-sided Wilcoxon rank-sum test. **g** Average ChIP-seq signal of BRD4 (GSE55062) around peak centers at AR unique peaks, AR and H3K27ac common peaks, and H3K27ac unique peaks. ±2 kb region of peak centers were binned by 50 bp windows. **h** Motif enrichment at H3K27ac peaks annotated to DHT-upregulated genes (704 peaks) or DHT-downregulated genes (336 peaks) at 2 h time point. H3K27ac peaks within ±20 kb region of gene TSS were selected. Source data are provided in Histogram and Immunoblot Source Data files.

expression of DHT-repressed genes that are associated with H3K27ac sites that do not overlap AR sites (H3K27ac only) decreased most significantly in response to DHT (Fig. 1f, right panel). DHT-repressed genes associated with H3K27ac/AR co-occupancy (H3K27ac and AR) were less decreased, and genes associated with AR-specific sites (AR only) or without AR or H3K27ac, were least decreased. As a comparison, expression of DHT-stimulated genes was most increased on genes with H3K27ac/AR co-occupancy (Fig. 1f, left panel). These analyses further indicate that a substantial portion of AR transcriptional repression function is indirect.

One mechanism that could contribute to indirect transcriptional repression is redistribution of transcriptional co-factors such as BRD4. Indeed, global analyses demonstrated that DHT increased BRD4 binding at AR sites that were not associated with H3K27ac under basal conditions (AR unique peaks), but decreased its binding globally at H3K27ac sites that were not AR associated (H3K27ac unique peaks) (Figs. 1g, S1f). BRD4 binding at H3K27ac/AR common peaks was unchanged, which would be consistent with basal BRD4 binding at these sites prior to DHT stimulation, while the increased binding at AR unique peaks may reflect direct recruitment by AR and increases in

histone acetylation in response to the DHT at these sites. Moreover, BRD4 binding at DHT-repressed genes was decreased by DHT, while BRD4 binding at DHT-stimulated genes was increased by DHT (Fig. S1g). To determine whether the DHT-mediated decrease in BRD4 binding was affecting a functionally related set of genes, we identified the subset of DHT-repressed genes that had the greatest decrease in BRD4 binding (Fig. S1h). Interestingly, gene ontology biological process and canonical pathway enrichment analyses suggested involvement in regulation of apoptosis (Fig. S1i). In any case, these results support BRD4 redistribution as a mechanism contributing to AR-mediated transcriptional repression.

We also performed motif analyses around basal H3K27ac peaks in loci of genes that are stimulated versus repressed by DHT at the 2 h time point to determine whether AR may be altering recruitment of other transcription factors (TFs) at DHT-repressed genes. ETV6, ETS1, and RUNX2 motifs were enriched at H3K27ac sites in DHT-stimulated gene loci, while an enrichment of CBX3, JUN, HSFY2, and MYC/MAX motifs was identified in DHT-repressed loci (Figs. 1h, S1j). Of note, AR and FOXA1 motifs were not enriched at DHT-stimulated genes as this analysis was centered on H3K27ac peaks present prior to DHT-stimulation. Interestingly, the E2F1 motif was enriched at genes that were repressed after 2 h of DHT treatment, consistent with previous reports that AR could directly enhance recruitment of RB1 protein to E2F1 sites (Fig. S1k)[14,24].

**MYC is an AR-repressed transactivator that binds to DHT-repressed genes**. To determine whether expression of the above or other TFs was altered in response to DHT, we assessed mRNA for all 1875 human TFs in VCaP cells in response to DHT[25]. RNA-seq showed that MYC was amongst the most significantly decreased TF at all time points (Fig. 2a). AR mRNA was not changed at 2 h, but was significantly decreased at later time points, as reported previously[26]. This expression data in combination with the motif enrichment analyses showed that MYC is the only TF that passes all filters at both 2 and 24 h of DHT-stimulation (Fig. S2a). Similar global analyses of TF expression further demonstrated that MYC mRNA was substantially repressed after 24 h of DHT stimulation in LNCaP-AR cells, but not in the parental LNCaP cells (Fig. 2b, c).

MYC repression in VCaP cells at 24 h, but not LNCaP cells, occurred over a broad range of DHT concentrations (as low as 10 pM) (Fig. 2d, e). However, time-course studies indicated that MYC was weakly and transiently repressed by DHT in LNCaP cells (although this did not reach statistical significance) (Fig. 2f). This transient MYC repression is similar to what we have reported previously for a cell cycle gene subset, and appears to reflect a competing effect of DHT to transactivate other genes that drive cells through the G1/S checkpoint[14,24]. Significantly, the deeper and prolonged MYC repression in VCaP is related to higher AR levels, as MYC mRNA was similarly decreased upon DHT stimulation in LNCaP-AR cells (Fig. 2g). As expected, the DHT repressive effect in VCaP was attenuated by the AR antagonist enzalutamide and by AR siRNA (Fig. S2b, c).

DHT also caused a rapid decrease in MYC protein (Fig. 2h), which combined with the enrichment for the MYC motif at AR repressed genes, suggested that decreased MYC binding may be a basis for the DHT-mediated repression of a subset of genes. To test this hypothesis, we next performed MYC ChIP-seq in vehicle versus DHT-stimulated VCaP cells. Using the threshold of $q$ value < 0.05, 14,697 MYC-centralized peaks were called under vehicle condition, and this decreased to 8076 upon 4 h of DHT induction, with 2060 of these being new peaks (Figs. 2i, S2d, e). Notably, these persistent peaks after DHT reflected those with the greatest peak intensity prior to DHT, and were markedly decreased in intensity by DHT (Figs. 2j, S2e). Conversely, peak intensity for the 2060 new MYC peaks was low, and most were associated with weak MYC binding prior to DHT that did not substantially increase, but became significant primarily due to lower background after DHT treatment (Fig. S2e).

We then used BETA to determine whether MYC binding was associated with gene repression in response to DHT. Notably, basal MYC binding was significantly associated with genes that were DHT-repressed at all time points, with the association increasing over time (Fig. 2k). Interestingly, basal MYC binding was also associated with genes that were upregulated by DHT at the 10 and 24 h time points, suggesting that AR may be compensating for reduced MYC at these genes (see below). We also analyzed separately the MYC sites that were lost, gained, or persisted after DHT stimulation. As noted above, basal MYC binding was greatest at the sites that persisted (shared sites), and most markedly declined in response to DHT. Consistent with this, BETA showed that these shared sites were most strongly associated with DHT-repressed genes at all time points, with a weaker, but still significant, association for the other sites (Fig. S2f).

To identify the subset of genes whose expression was altered by reduced MYC (independently of DHT-stimulation), we then performed RNA-seq analyses in VCaP cells after MYC knockdown by siRNA (Fig. S2g, h). Annotation of the MYC ChIP-seq and RNA-seq data by the BETA tool showed that basal MYC binding was strongly associated with MYC-upregulated genes, but only weakly associated with MYC-downregulated genes, consistent with it acting primarily as a transcriptional activator (Fig. 2l). As expected, the top KEGG pathways and GO cellular component terms enriched in the gene set that was decreased by siMYC (MYC-Up) were related to cell cycle and DNA damage repair, and a subset of these were also enriched in the DHT-repressed gene set (AR_Down) (Fig. S2i, j). Analysis of a previous study in VCaP cells showed the potent synthetic nonmetabolized AR ligand R1881 suppressed the expression of more genes than DHT, and correspondingly decreased more MYC regulated genes, which were similarly enriched in cell cycle and DNA repair-related pathways (Fig. S2k, l).

We then determined the overlap between genes that were decreased by MYC siRNA (MYC_upregulated) and down-regulated by DHT (DHT_downregulated), which indicated that reduced MYC could account for ~8% of DHT-repressed genes (45 of 567 and 49 of 593 DHT-repressed genes at 10 and 24 h, respectively) (Fig. 2m). Conversely, this group of overlapping genes reflected ~20% of the genes that were decreased by MYC siRNA. This latter overlap with genes repressed by MYC siRNA may be less than expected given the marked DHT-mediated decrease in MYC. However, this ~20% may reflect genes that are most acutely (within 24 h) altered by decreased MYC (and not directly or indirectly stimulated by AR), versus those that are decreased by the siRNA-mediated decrease in MYC over 2 days. GO biological process terms enriched in this common gene set of MYC-stimulated and DHT-repressed genes were pseudouridine synthesis and mRNA modification (Fig. S2m). Together, these results support MYC downregulation as a mechanism that contributes to DHT-mediated transcriptional repression.

**MYC acts cooperatively with AR at subset of DHT-stimulated genes**. The data above are consistent with the hypothesis that reduced MYC binding is a contributor to DHT-mediated transcriptional repression. However, basal MYC binding was also associated with DHT-stimulated genes (see Fig. 2k), and the basis for this association was less clear. Significantly, while peak-gene proximity analysis (within 20 Kb of transcriptional start sites) in

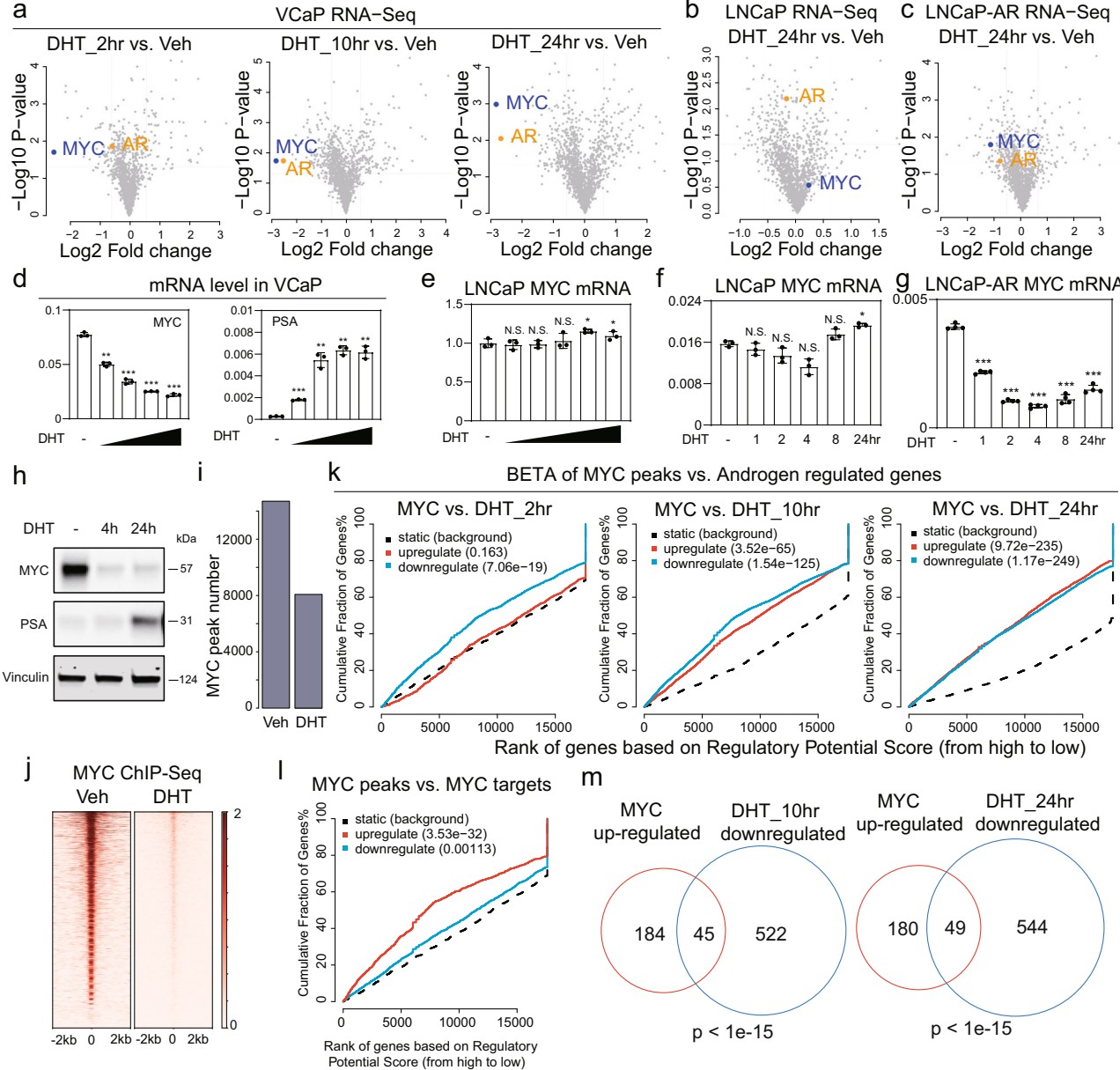

**Fig. 2 MYC is a DHT-repressed TF that binds to androgen repressed genes.** Volcano plots showing changes of TF genes ($n = 1875$) in DHT-treated VCaP cells (**a**), LNCaP cells (**b**), and LNCaP-AR cells (**c**). MYC and AR messages were highlighted. **d** RT-qPCR for MYC and PSA mRNA (normalized to GAPDH) in DHT-stimulated VCaP cells (10 pM, 100 pM, 1 nM, and 10 nM) for 24 h. **e** RT-qPCR analysis of MYC mRNA expression (normalized to GAPDH) in LNCaP cells under a range of DHT doses (10 pM, 100 pM, 1 nM, 10 nM, 100 nM) for 24 h treatment. RT-qPCR analysis of MYC mRNA expression (normalized to GAPDH) in LNCaP (**f**) and LNCaP-AR (**g**) cells under 10 nM DHT treatment for indicated times. For **d–g**, data are presented as mean values ± SD for three biologically independent samples. $P$ values were by two-sided Student's $t$ test. N.S., not significant; *, $P < 0.05$; **, $P < 0.01$; ***, $P < 0.001$. **h** Western blotting of MYC and PSA in VCaP cells with 10 nM of DHT for indicated time points. Vinculin is loading control (representative of three independent experiments). **i** Barplot showing the total peak number of MYC ChIP-seq in VCaP cells grown in androgen depleted medium and then stimulated with vehicle or DHT (10 nM, 4 h); cutoff: $q$ value < 0.05. **j** Heatmap showing MYC binding signal in VCaP cells without or with DHT (10 nM, 4 h). **k** BETA to assess the association between MYC binding (under vehicle condition) and androgen-regulated gene expression in VCaP cells. The red line indicates MYC-upregulated genes and the light blue line indicates the MYC-downregulated genes. **l** BETA to determine the association between MYC binding (under vehicle condition) and MYC-regulated gene expression in VCaP cells. The red line indicates MYC-upregulated genes and the light blue line indicates the MYC-downregulated genes. **m** Overlap between MYC positively regulated genes (fold change > 2 and $P < 0.05$) and DHT-repressed genes (fold change < 1/2 and $P < 0.05$). $P$ values were determined by a hypergeometric test in **m**. Source data are provided in Histogram and Immunoblot Source Data files.

VCaP cells showed that basal MYC binding was highly enriched amongst DHT-repressed genes (Odds Ratio 3.13), MYC binding was also enriched (although lower) amongst DHT-stimulated genes (Odds Ratio 2.06) (Fig. 3a). Conversely, as expected, AR binding was highly associated with DHT-stimulated genes.

This MYC association with DHT-stimulated genes suggested that a subset of these genes may be co-regulated by MYC and AR. Consistent with this hypothesis, comparison of MYC and AR ChIP-seq data showed that ~25% of MYC binding sites in vehicle-treated VCaP cells overlapped AR binding sites (3716 of

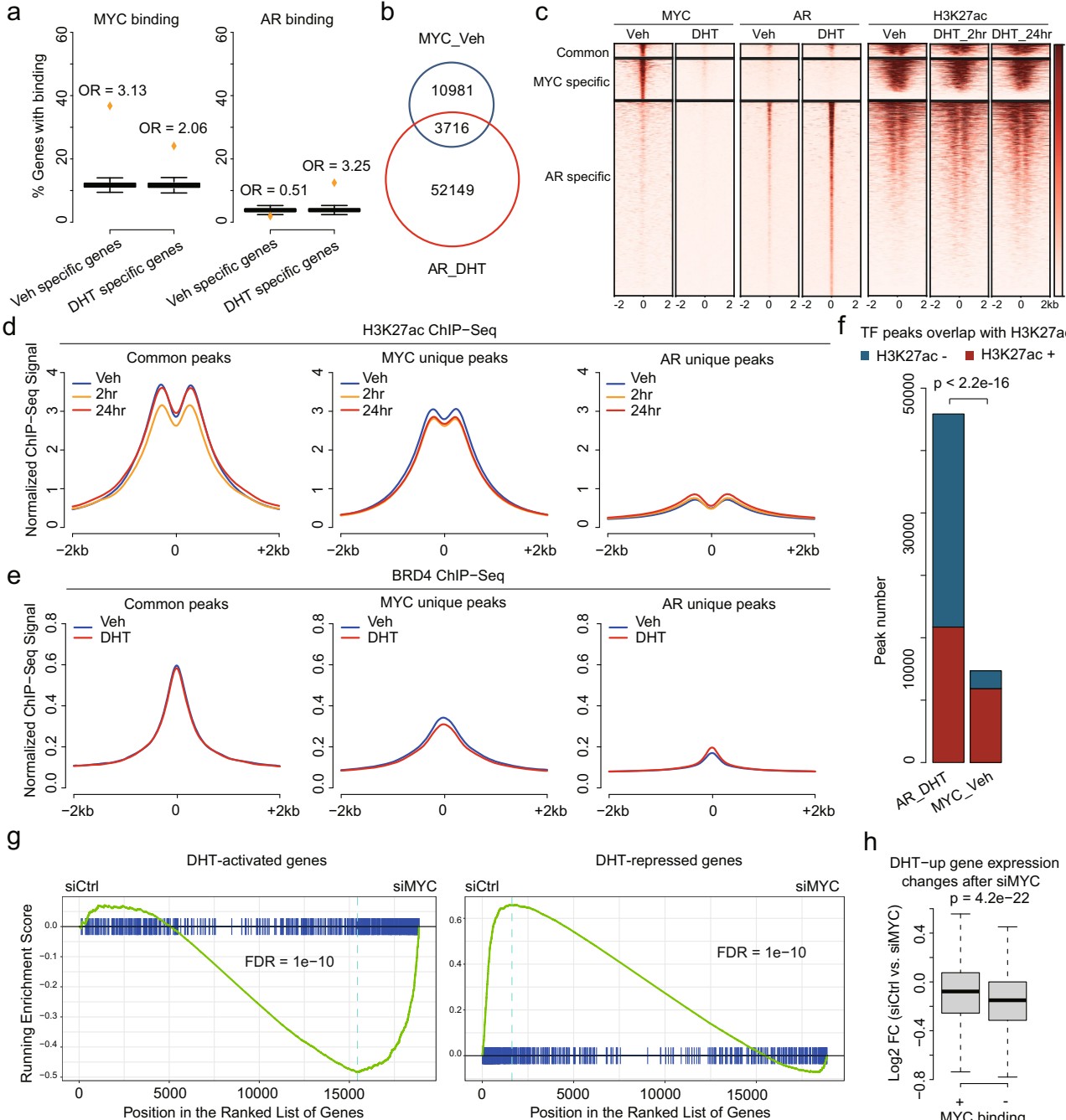

**Fig. 3 Functional interaction between MYC and AR proteins. a** Enrichment analysis of vehicle-specific or DHT-specific genes with MYC or AR ChIP-seq peaks in proximity to (±20 kb) the TSSs. Orange points represent actual vehicle-specific or DHT-specific gene sets. Boxes represent 1,000 times random sampling from the whole transcriptome. Same number of genes as in the vehicle-specific and DHT-specific gene sets were used for random sampling, respectively. Odds ratio (OR) was calculated by comparing the percentage between actual gene sets and the average of random sampling. From the left to the right, $P < 2.2\text{e-}16$, $P < 2.2\text{e-}16$, $P = 1$ and $P < 2.2\text{e-}16$, respectively, by one-sided Student's $t$ test. Box limits, 1st and 3rd quartiles; whiskers, 1.5× inter-quartile range; center line, median. From the left to right, $n = 1292, 1212, 1292$, and 1212, respectively. **b** Venn diagram showing the overlap between MYC ChIP-seq peaks under vehicle condition and AR ChIP-seq peaks under DHT condition in VCaP cells. The MYC peaks overlapping with AR peaks are counted as common peaks. **c** Heatmaps showing the ChIP-seq signal of MYC, AR, and H3K27ac at MYC/AR common peaks, MYC-specific peaks, and AR-specific peaks in VCaP cells with or without DHT. **d** Pileup of H3K27ac ChIP-seq signal at MYC/AR common peaks, MYC-specific peaks, and AR-specific peaks in VCaP cells with or without DHT. **e** Pileup of BRD4 ChIP-seq signal at MYC/AR common peaks, MYC-specific peaks and AR-specific peaks in VCaP cells with or without DHT. **f** Barplot showing the number of liganded-AR and MYC peaks overlapping with enhancers (as marked by H3K27ac) in VCaP. The $P$ value was determined by Chi-squared test. **g** Gene Set Enrichment Analysis (GSEA) using DHT-activated genes and DHT-repressed genes as gene sets to determine whether these two gene sets were regulated by MYC in complete FBS medium. **h** Boxplot showing log2 fold changes (siCtrl vs. siMYC) in complete FBS medium of DHT-upregulated genes. These genes were separated into two groups based on MYC binding status. Box limits, 1st and 3rd quartiles; whiskers, 1.5× inter-quartile range; center line, median. From the left to right, $n = 392$ and 216, respectively. $P$ value was determined by two-sided Wilcoxon rank-sum test.

14,697 sites) (Fig. 3b). Conversely, only ~7% of AR peaks after DHT stimulation overlapped sites with basal MYC binding. As expected, motifs for MYC, AR, and FOXA1 were enriched at the AR/MYC common peaks, while only the MYC or AR and FOXA1 motifs were enriched at the MYC unique or AR unique peaks, respectively (Fig. S3a). Notably, HOXB13 and GATA2 motifs were enriched at AR unique peaks, but not at AR/MYC common peaks. Interestingly, the ERG motif was enriched at the common and unique sites.

MYC binding at these 3716 common AR/MYC sites, as well as at MYC specific sites, was markedly decreased by DHT (Fig. 3c). We then characterized the regulatory dynamics at MYC and AR binding sites by comparing the time-dependent effects of DHT on H3K27ac signal at these sites. For MYC unique peaks, H3K27ac signal was decreased after 2 and 24 h of DHT treatment, consistent with reduced MYC at these sites (Figs. 3c, d, and S3b). In contrast, H3K27ac at AR unique peaks was elevated by DHT at 2 and 24 h, although basal H3K27ac at these sites was low and the increase was modest. This is consistent with previous data showing that a large fraction of AR binding sites detected by ChIP are in regions of closed chromatin and lack H3K27ac[27].

Notably, for AR and MYC common sites, baseline H3K27ac was high and showed a transient decrease at 2 h of DHT stimulation, but was restored to baseline at the 24 h time point (Figs. 3c, d, and S3b). Consistent with this result, BRD4 binding at AR/MYC common peaks was unchanged at 24 h after DHT stimulation, and was decreased at MYC unique sites (although this latter loss may reflect both decreased MYC binding at these sites and redistribution of BRD4 to AR sites) (Figs. 3e and S3c). Together, these findings support reduced MYC, as well as cofactor redistribution, as a mechanism for DHT-mediated transcriptional repression at genes regulated specifically by MYC (MYC unique genes). Moreover, they indicate that genes with both MYC and AR binding sites (common peaks) are dynamically co-regulated by these TFs, with an initial rapid decrease in activity due to reduced MYC, followed by compensation through AR.

As noted above, relative to AR unique sites, AR/MYC common binding sites, as well as MYC unique sites, are associated with high H3K27ac (Fig. 3f). This in part reflects greater association of AR/MYC common versus AR unique sites with promoters (Fig. S3d), but in any case, indicates that MYC may amplify AR transcriptional activity. Indeed, only 2.6% of 55,865 AR unique binding sites were associated with DHT-altered genes at 24 h, versus 7.1% of 3716 AR/MYC shared sites. Moreover, a large fraction of genes that are modulated by DHT that do not have shared AR/MYC sites do nonetheless have MYC unique sites, so that MYC binding is enriched for both genes that are DHT upregulated and downregulated (Fig. S3e). GO term analysis further shows substantial functional overlap between genes with AR/MYC common versus MYC unique genes (Fig. S3f).

As expected, GSEA showed that the set of genes that were decreased after MYC depletion was enriched for DHT-repressed genes, consistent with MYC downregulation contributing to the DHT-repressed genes (Fig. 3g). Surprisingly, genes that were increased after MYC depletion were enriched for DHT-stimulated genes (Fig. 3g). Significantly, this increase was greater for AR regulated genes without MYC binding sites, indicating that it does not reflect a direct repressive function of MYC, and similarly to the indirect effects of DHT, may reflect cofactor redistribution to AR regulated genes that are mitigated by reduced MYC at AR/MYC co-regulated genes (Fig. 3h).

Analysis of a previously reported data set (GSE82223) confirmed that the MYC-activated gene set was repressed by DHT in VCaP cells, and this repression was prevented by treatment with AR siRNA (Fig. S3g, h). In contrast, the MYC-activated gene set in LNCaP cells was enriched by DHT-stimulation, and this also was lost upon treatment with AR siRNA. This is consistent with the combined stimulatory effects of AR and persistence of MYC in DHT-stimulated LNCaP cells, versus decreases of MYC in VCaP cells.

**AR-mediated transactivation at MYC independent genes is repressed by MYC overexpression.** The above studies indicated that AR and MYC cooperate directly to regulate a subset of genes, and may cooperate indirectly through cofactor redistribution. To address this further, we examined LA cells (LNCaP cells over-expressing AR) and LAM cells (LNCaP cells overexpressing AR and MYC), which were maintained in medium containing basal androgen (FBS medium) (Fig. 4a). We first assessed effects of this MYC overexpression on genes regulated by AR alone (KLK3, TMPRSS2) versus regulated by MYC alone (RAD51AP1, LMNB1) or with shared AR and MYC binding sites (NKX3-1) (Fig. S4a–c). As expected, DHT did not greatly suppress the overexpressed MYC in the LAM cells (Fig. 4b). MYC overexpression markedly suppressed KLK3 and TMPRSS2 expression, but not NKX3-1 expression. Moreover, it prevented the DHT-mediated decrease in LMNB1 and RAD51AP1, confirming that suppression of these genes by DHT is due to MYC downregulation (Fig. 4c–e). Next, we globally assessed AR and MYC-regulated genes. As expected, MYC overexpression increased expression of genes that are downregulated by MYC siRNA (Fig. 4f, left panel). Significantly, MYC overexpression decreased expression of DHT upregulated genes, consistent with coactivator redistribution, and increased the expression of DHT downregulated genes, a subset of which are MYC regulated (Fig. 4f, middle and right panels). We carried a similar analysis using data from a previous study comparing gene expression in LNCaP cells overexpressing MYC versus parental LNCaP[19], which showed a comparable marked decrease in AR regulated genes in the MYC overexpressing LNCaP cells (Fig. S4d).

We then specifically examined MYC effects on AR binding at the enhancers of two strongly AR-regulated genes, KLK3 and TMPRSS2, that were repressed in the LAM versus LA cells (Fig. 4g). By ChIP we found decreased H3K27ac in the LAM cells, but only modest or no decrease in AR binding, supporting a decrease in coactivator recruitment (Fig. 4h). These results are consistent with a previous study that found MYC overexpression could decrease AR transcriptional activity[19]. We next examined the effects of depleting MYC on expression of these genes and their enhancers in VCaP cells. MYC siRNA did not increase AR, but markedly increased KLK3 and TMPRSS2 mRNA (Fig. 4i, j). ChIP similarly showed no increase in AR binding, but an increase in H3K27ac at the KLK3 and TMPRSS2 enhancers, consistent with increased coactivator recruitment (Fig. 4k).

**MYC pathway is activated after castration.** Since DHT-stimulation suppresses MYC, we speculated that androgen deprivation would do the reverse and lead to MYC activation at the expense of the AR pathway. To test this, we cultured VCaP cells in DHT-supplemented medium (FBS with 10 nM DHT, D), and then shifted them to FBS medium without adding DHT (vehicle, V) or FBS medium with the AR antagonist enzalutamide (ENZ, E). Both removal of the DHT and addition of ENZ induced MYC expression (Fig. 5a, upper panel) and conversely, as expected, decreased PSA (Fig. 5a, lower panel). We also examined the effects of bicalutamide (B), another AR antagonist that impairs coactivator binding, but in contrast to ENZ more robustly increases AR nuclear translocation. Similar to ENZ, treatment with bicalutamide did not suppress MYC expression in androgen starved cells, or in cells cultured in full FBS medium

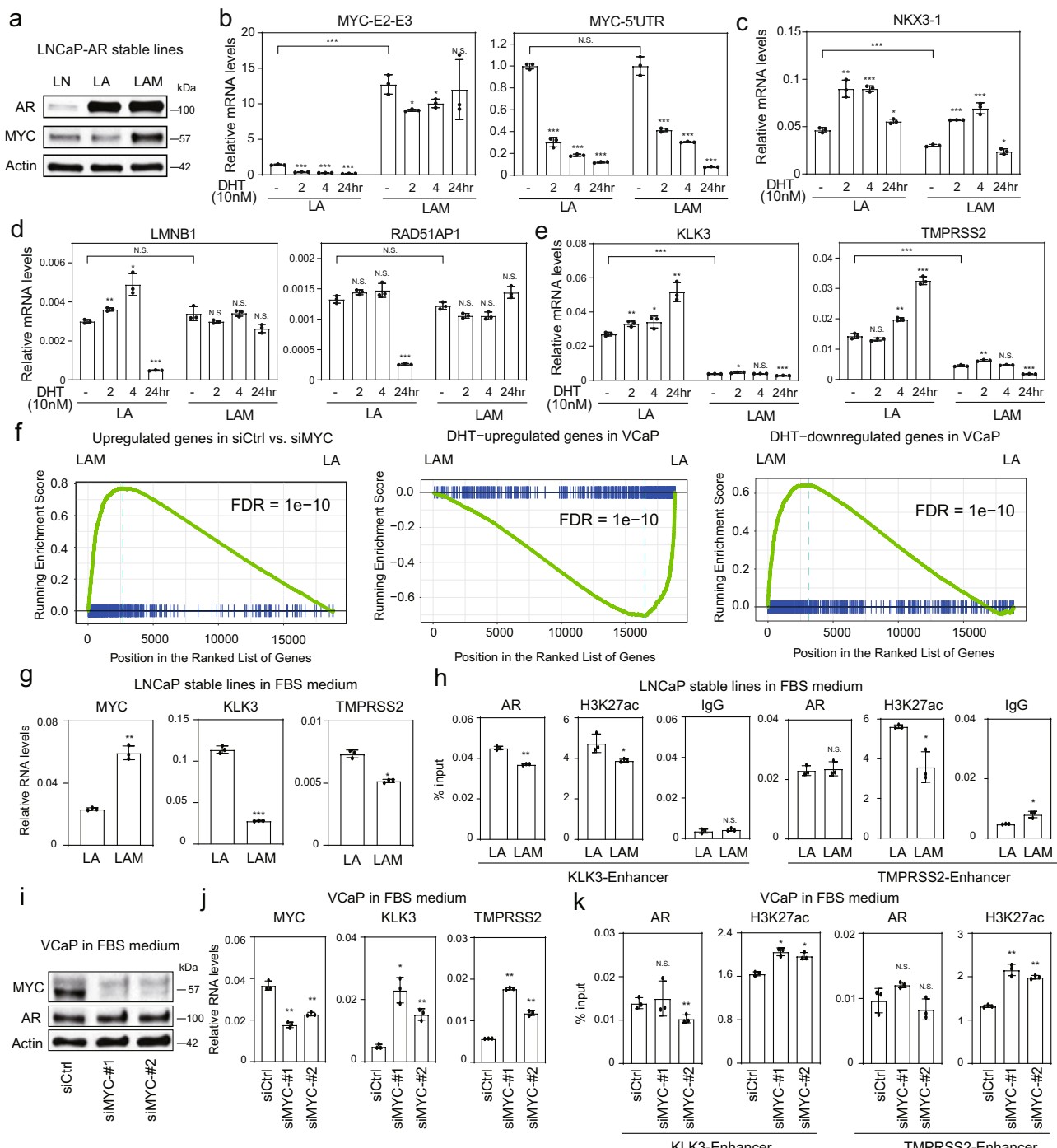

(Fig. S5a, b). We also assessed a novel AR degrader (ARCC-32, CC) that targets the AR for ubiquitylation and degradation[28], which similarly induced MYC expression at both mRNA and protein levels (Fig. 5b, c).

Next, we globally assessed the effects of AR blockade with ENZ on expression of all TFs, and identified MYC and AR as among the most significantly induced TFs (Fig. 5d). Furthermore, ENZ-elicited gene expression showed a strong negative correlation with the response to DHT (as expected), and a positive correlation with genes that are increased in control versus MYC depleted cells (Fig. 5e), while DHT-stimulated genes had a negative correlation with MYC stimulated genes (Fig. S5c). Consistent with these transcriptional effects, ENZ also elicited changes in H3K27ac, which was increased at MYC-specific peaks, decreased

at AR-specific peaks, and unchanged at the AR/MYC shared peaks, consistent with increased MYC compensating for the loss of AR activity at shared sites (Figs. 5f and S5d). In addition, KEGG pathway enrichment analysis demonstrated a marked overlap between pathways that were increased by ENZ and activated by MYC, including pathways related to DNA replication, RNA transport, and ribosome biogenesis (Fig. 5g). A similar profiling transition was noted in GO enrichment analysis (Fig. S5e).

To determine whether these effects also occurred in vivo, we established VCaP xenografts in intact male mice, which were then subjected to castration. Analysis at 4 days and 3 weeks after castration showed a rapid and persistent increase in MYC staining and, as expected, decreased PSA staining (Fig. 5h).

**Fig. 4 MYC alters androgen-responsive transcriptome by mediating co-factor redistribution. a** Western blot analysis of AR or AR/MYC overexpression in LNCaP cells. LN, LNCaP; LA, LNCaP-AR; LAM, LNCaP-AR-MYC. The experiment was repeated independently three times with similar results. LNCaP-AR (LA, for AR overexpression) and LNCaP-AR-MYC (LAM, for AR and MYC overexpression) were cultured in androgen-depleted medium and treated with DHT for indicated time-points. Total RNA was subjected to real-time RT-PCR analyses. MYC-5′-UTR was based on primers targeting MYC-5′ UTR region; MYC-E2-E3 was based on primers spanning MYC exon-2 and exon-3 regions (**b**). The relative expression of MYC and AR targets was also determined by RT-qPCR (**c–e**). **f** Gene Set Enrichment Analysis (GSEA) using upregulated genes in siCtrl vs. siMYC (in complete FBS medium), DHT-activated genes, and androgen-repressed genes as gene sets (all from VCaP cells) to determine whether these two gene sets were regulated by MYC overexpression in LNCaP-AR cells. **g** RT-qPCR to determine MYC, KLK3, and TMPRSS2 mRNA levels in LA and LAM cells in full FBS medium. **h** ChIP-qPCR to determine AR and H3K27ac signal at KLK3 and TMPRSS2 enhancers in LA and LAM cells within full FBS medium. **i** Western blot analysis of MYC knockdown in VCaP cells. The experiment was repeated independently three times with similar results. **j** RT-qPCR to determine MYC, KLK3, and TMPRSS2 mRNA levels in VCaP cells in full FBS medium. **k** ChIP-qPCR to determine AR and H3K27ac signal at KLK3 and TMPRSS2 enhancers in VCaP cells in full FBS medium. For **b–e**, **g**, **h**, **j**, and **k**, data are presented as mean values ± SD; for each test, $n = 3$ biologically independent samples. For **b–e**, $P$ values were determined by Wilcoxon Rank Sum Test and the multiple tests were further corrected by Bonferroni-Holm method; N.S., not significant; *, adjusted $P < 0.05$; **, adjusted $P < 0.01$; ***, adjusted $P < 0.001$. For **g**, **h**, **j**, and **k**, $P$ values were calculated by two-sided Student's $t$ test; N.S., not significant; *, $P < 0.05$; **, $P < 0.01$; ***, $P < 0.001$. Source data are provided in Histogram and Immunoblot Source Data files.

Consistent with this result, RNA-seq analysis showed marked and rapid increases in MYC and AR mRNA in response to castration (Fig. S5f). Conversely, we also addressed whether androgen treatment in castration-resistant tumors in vivo would decrease MYC. For this purpose, mice bearing VCaP xenografts were castrated and observed until tumor progression. Mice were then treated with supraphysiological androgen (testosterone, T), which we showed previously caused VCaP tumor regression[26]. This markedly activated AR-dependent transcription (as assessed by PSA mRNA), but substantially repressed MYC, and both effects were effectively reversed by ENZ co-treatment (Fig. 5i).

We next addressed whether androgen deprivation therapy and subsequent decreased AR activity may be increasing MYC in clinical samples. For this analysis, we compared gene expression in primary untreated PCa versus samples from men who had progressed after androgen deprivation therapy (castration-resistant prostate cancer, CRPC). As *MYC* and *AR* gene amplification are common in CRPC, we excluded these cases from our analysis. The expression of MYC is slightly increased in primary PCa (as compared to normal), but is significantly higher in CRPC, and MYC signature genes are also dramatically overexpressed in CRPC as compared with primary PCa (Fig. 5j). Consistent with this finding, H3K27ac at MYC sites (MYC specific and MYC/AR common) was increased in primary PCa clinical samples versus normal, and further markedly increased in PDXs from CRPC (Fig. 5k). Finally, CRPC samples showed an increase in expression of MYC signature genes with both AR/MYC co-binding and MYC unique sites (Fig. S5g). Together these in vitro and in vivo results indicated that loss of AR activity in response to androgen deprivation at AR/MYC co-regulated genes is buffered by an increase in MYC, along with an increase in expression of MYC-unique genes that may further enhance tumor growth.

**Locus-wide repression of the 8q24 TAD by DHT correlates with the decline in distal enhancer interaction.** The *MYC* gene is embedded in a topologically associated domain (TAD) containing multiple distal enhancers that engage with the *MYC* promoter in a tissue-specific manner[29]. The *MYC* promoter in PCa cells has been found to interact with a centromeric prostate-specific super-enhancer (SE) overlapping *PCAT1*[4,5]. Global analysis of SE distribution in VCaP cells, based on H3K27ac, confirmed that the region spanning *PCAT1-PRNCR1* was the top-ranked amongst all 1244 SEs, and showed that the telomeric *PVT1* region TAD contains a weaker SE (Fig. 6a). Consistent with this second SE, LNCaP Hi-C data shows that this locus contains sub-TADs on the left and right arms of *MYC* (Fig. S6a). The *PCAT1* SE has three smaller sub-SEs, and H3K27ac at each site

and overall was markedly decreased upon DHT stimulation (Figs. 6a, b, S6a, b). In contrast, the SE in the right arm *PVT1* region was not clearly perturbed by DHT.

The *PCAT1* region has been denoted as PCa risk region 2 (RR2). Its PCa-specific interaction with the *MYC* gene promoter has been demonstrated in LNCaP cells[4], and CRISPR/Cas9-directed knock-out of the *PCAT1* SE decreases MYC gene expression in VCaP cells[30]. The 8q24 TAD also harbors lncRNAs that have been implicated in *MYC* regulation and its oncogenicity[31]. Therefore, we explored whether these were modulated by AR similarly to MYC. Indeed, RNA-seq analysis showed that DHT treatment in VCaP cells led to reduction of transcripts in the entire 8q24 locus (Fig. S7a). Analysis by qRT-PCR confirmed that DHT substantially decreased *PCAT1* and *PVT1* expression in VCaP cells (Fig. S7b). Two additional lncRNAs (CCAT1 and CASC8) in this locus were also androgen-repressed. Expression of *PCAT1* and *PVT1* were similarly suppressed by androgen in LNCaP-AR cells (Fig. S7c). In contrast, in LNCaP cells DHT only transiently decreased expression of these genes (Fig. S7b). Finally, in VCaP cells adapted to androgen-supplemented medium, both androgen deprivation and ENZ treatment led to increases in *PCAT1* and *PVT1* expression, similar to the response of *MYC* (Fig. S7d).

In contrast to these genes in the 8q24 TAD, three genes (two coding genes, *FAM84B* and *GSDMC*, and one lncRNA gene *LINC00977*) on the left and right boundaries of 8q24 locus were not repressed, or were only modestly transiently repressed, by DHT (Fig. S7e). These genes are not linked to the overall TAD architecture within the *PCAT1-PVT1* region (Fig. S6a). Using VCaP siAR data, we also confirmed that the DHT repressive effects on *PCAT1* and *PVT1* were mediated by AR (Fig. S7f). Finally, similarly to MYC, expression of PCAT1 and PVT1 are increased in CRPC, while expression of directly AR regulated genes such as KLK3 and TMPRSS2 is decreased (despite the marked increase in AR mRNA) (Fig. S7g). Together these findings indicate that the entire 8q24 TAD is interwoven into a locus-wide regulatory network, and that regulation of this locus by androgen has clinical relevance in PCa.

**8q24 super-enhancer driving *MYC* transcription is disrupted by DHT.** Consistent with the above results, H3K27ac HiChIP showed that the *PCAT1/MYC* region had the highest interaction frequency across chr8 under vehicle condition, and had the largest decline upon DHT stimulation (Fig. S8a). These *PCAT1* SE looping interactions seen by H3K27ac HiChIP were focused on the *MYC* promoter, and both the *MYC* promoter and *PCAT1* SE had additional looping interactions with lncRNA genes across the

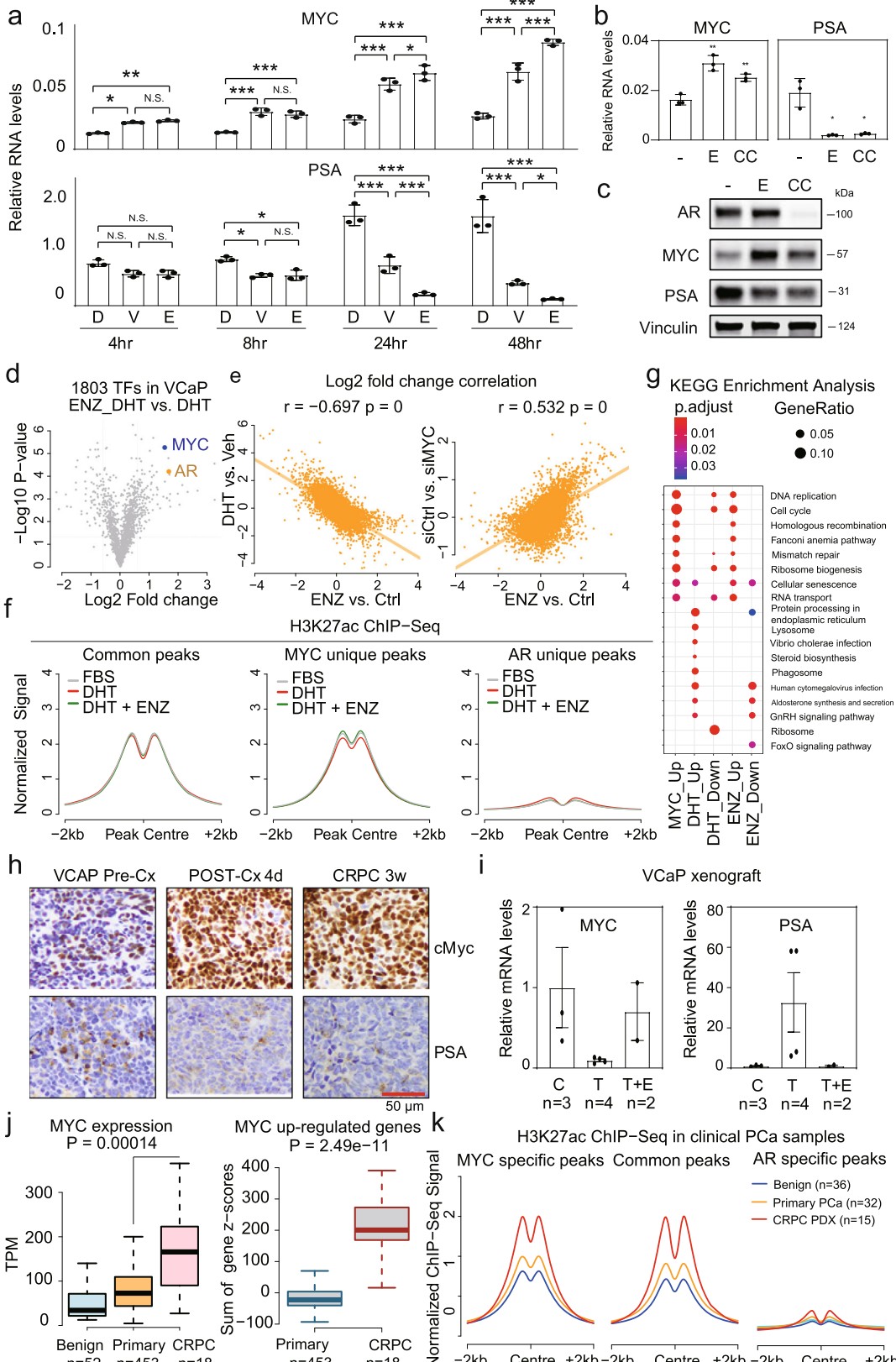

8q24 TAD (Figs. 6c and S8b), indicating a locus-wide interaction network.

Significantly, the *PCAT1* SE interaction with the *MYC* promoter, as detected by H3K27ac HiChIP, was greatly decreased by DHT (Fig. 6c). To determine whether this reflected decreased interaction versus just loss of H3K27ac, we used circular

chromosome conformation capture (4C) to directly examine *MYC* promoter interactions (Fig. S8c). Consistent with the H3K27ac HiChIP, this showed *MYC* promoter interaction with multiple sites localized within the 8q24 TAD, including robust interaction peaks within the *PCAT1* SE (Figs. 6c, S8d, e). Moreover, the latter *PCAT1* SE interactions were markedly

**Fig. 5 MYC signature is activated upon immediate castration in compensation of AR pathway. a** VCaP cells in androgen-proficient medium (10% FBS + 10 nM DHT, or D) and then DHT removal (10% FBS + vehicle, or V) or anti-androgen treatment (10% FBS + 10 μM ENZ, or E) for indicated time points were assessed by qRT-PCR. Data are mean values ± SD for three biologically independent samples. *P* values were determined by one-way ANOVA. N.S., not significant; *, *P* < 0.05; **, *P* < 0.01; ***, <0.001. VCaP cells in androgen-proficient medium (10% FBS + 10 nM of DHT) and were treated with enzalutamide (10% FBS + 10 μM of ENZ, E) or an AR degrader ARCC-32 (CC, 500 nM) for 24 hrs and assessed by qRT-PCR (**b**) and Western blotting (**c**). Data in **b** are mean ± SD of three independent samples; *P* values were determined by two-sided Student's *t* test. N.S., not significant; *, *P* < 0.05; **, *P* < 0.01. Blot in **c** is representative of three experiments. **d** Volcano plot showing expression changes of all human TF genes caused by ENZ in VCaP cells. **e** Left: reverse correlation between ENZ-elicited and DHT-elicited expression changes; Right: correlation between expression changes in response to ENZ versus MYC knockdown. **f** H3K27ac ChIP-seq signal at MYC/AR common peaks, MYC-specific peaks, and AR-specific peaks in VCaP cells in FBS medium ± DHT or DHT + ENZ. **g** KEGG pathway enrichment analysis of MYC-regulated genes, DHT-regulated genes, and ENZ-regulated genes. Pathways of MYC-downregulated genes were below the threshold. **h** VCaP xenografts precastration, 4 days postcastration, and at relapse (CRPC 3 weeks) analyzed by IHC for MYC and PSA. Data are representative of 3 mice at the 4 day postcastration point, and 5 mice at the other times. **i** VCaP xenografts that relapsed after castration were sacrificed (C, 3 mice) or were treated for 2 days with testosterone ± ENZ (T, 4 mice or T + E, 2 mice), and assessed by qRT-PCR for MYC and PSA. Data are mean ± SE for the 2–4 mice. **j** MYC expression and z-scores for MYC-upregulated genes in clinical benign (TCGA), primary PCa (TCGA), and CRPC (patients who relapsed after ADT) RNA-seq data (phs001648.v1.p1). Samples with *AR* or *MYC* amplification were excluded. Box limits, 1st and 3rd quartiles; whiskers, 1.5× inter-quartile range; center line, median. From the left to right, *n* = 52, 453, 18, 453, and 18, respectively. *P* values were determined by two-sided Wilcoxon rank-sum test. **k** H3K27ac ChIP-seq for MYC/AR common peaks, MYC-specific peaks, and AR-specific peaks in clinical samples from GSE130408. Source data are provided in Histogram and Immunoblot Source Data files.

---

attenuated by DHT, while interaction with sites in *PVT1* were retained or enhanced. These findings were further confirmed by chromosome conformation capture (3C) (Fig. 6c). These results show that DHT-mediated repression of *MYC* expression is associated with a robust architectural change in 8q24 TAD, with a marked decrease in *MYC* promoter interaction with the *PCAT1* SE, which may be partially compensated by an increased interaction with the *PVT1* region. Consistent with these findings, a recent report found that the *MYC* promoter region contains an enhancer docking site that can mount on tissue-specific enhancers from both arms in the 8q24 TAD[32]. Interestingly, a previous study found that *MYC* promoter interactions with enhancers in the *PVT1* locus could be enhanced by a distinct mechanism, inactivation of the *PVT1* promoter, in breast cancer cells[33,34].

**Convergence of dual AR regulatory functions in 8q24 *MYC* locus.** Although the above findings demonstrated that DHT disrupts the *PCAT1* SE interaction with *MYC*, this may be direct or indirect, and previous studies indicate that AR may have a transcriptional activation function at this site in some contexts[4,30,35,36]. Indeed, there is a positive correlation between AR and MYC mRNA in primary PCa, but not in benign prostate (Fig. S9a). Taking advantage of recent PCa clinical datasets[9], we next assessed AR binding to the *PCAT1* SE during PCa development and progression to CRPC. Averaging AR signals across samples showed gains in AR binding with PCa progression from normal (N) to primary (T) and then to metastatic CRPC (M) (Fig. 7a). These are associated with binding of FOXA1 and HOXB13, and with increased H3K27ac, consistent with AR playing a role in activation of this SE. Of note, the increases in CRPC may in part reflect genomic amplification of this region, although the disproportionate increase in H3K27ac suggests this area gains enhancer activity in CRPC. By AR HiChIP we confirmed that AR in the PCAT1 SE loops prominently to the *MYC* promoter, and that this interaction is decreased by androgen (Fig. 7b).

Assessment of chromatin accessibility by ATAC-seq also showed that the *PCAT1* gene is only accessible in PCa (Fig. S9b). Further evidence supporting a direct stimulatory effect of AR was obtained from ENCODE H3K27ac ChIP-seq datasets on cell lines, which showed that only the two AR-positive PCa cell lines (LNCaP and VCaP), and not two AR negative PCa cell lines (PC3 and DU145), had robust H3K27ac signals over the *PCAT1* gene, and that these were in alignment with AR tracks (Fig. S9c).

Moreover, AR binding sites in the *PCAT1* SE overlap with BRD4 sites, consistent with AR contributing to BRD4 binding (Fig. S9c). Interestingly, MYC is also associated with the major AR/BRD4 site. Finally, analysis of DNA methylation data at this overlapping AR/BRD4/MYC site showed a decrease in PCa versus adjacent normal tissue[37], although in a second study decreased methylation was found in both tumor and benign tissue (Fig. S9d)[38].

DHT substantially depleted H3K27ac at the *PCAT1* SEs in VCaP, but had less impact in LNCaP, consistent with the DHT repressive effects on MYC in AR-high PCa cells (Fig. S9c). We next addressed the basis for disruption of *MYC* enhancer-promoter interactions in response to DHT. One possible mechanism is redistribution of transcriptional coactivators, and in particular BRD4, which we showed is globally redistributed in response to DHT (see Fig. 1g). In support of this mechanism, ChIP-seq showed that BRD4 binding to the *PCAT1* SE was decreased by DHT at 24 h (Fig. S9c). We confirmed by ChIP-qPCR that DHT treatment caused a decrease in H3K27ac and BRD4 at the AR/MYC site that was comparable to or greater than the decrease by the direct BRD4 inhibitor JQ1 (Fig. 7c). Surprisingly, JQ1 did not have a substantial effect on BRD4 binding to this site, although it did cause a rapid decrease in MYC mRNA (within 20 min), which persisted for at least 24 h (Fig. 7d). This may possibly reflect the combined effects of BRD4 loss at multiple sites in the MYC enhancer. Notably, although decreases in BRD4 binding can substantially account for the suppression of MYC by DHT, DHT combined with JQ1 was more suppressive than JQ1 alone, indicating that DHT may be acting through additional cofactors.

Previous GRO-seq data confirmed that agonist liganded AR was rapidly repressing *MYC* transcription (Fig. S10a). To further assess the basis for this repression, we generated LNCaP stable lines overexpressing a panel of AR mutants that are deficient in dimerization, DNA binding, or nuclear localization. Among these mutants, C619Y (M2) is defective in DNA binding but not nuclear localization, while K630, 632, 633A (M4) is defective in both nuclear localization and transactivation[39]. Also generated were AR R598A-N599A (M1) with perturbed dimerization function, and C562, 595A (M3) that would entirely lose DNA binding and transactivation capacities (Fig. 7e). The wild-type (WT) and mutant ARs in these stable lines were overexpressed to similar levels (Fig. 7f). The DNA binding defective M1, M2, and M3 mutants markedly suppressed expression of PSA and TMPRSS2, consistent with their competing for binding of nuclear

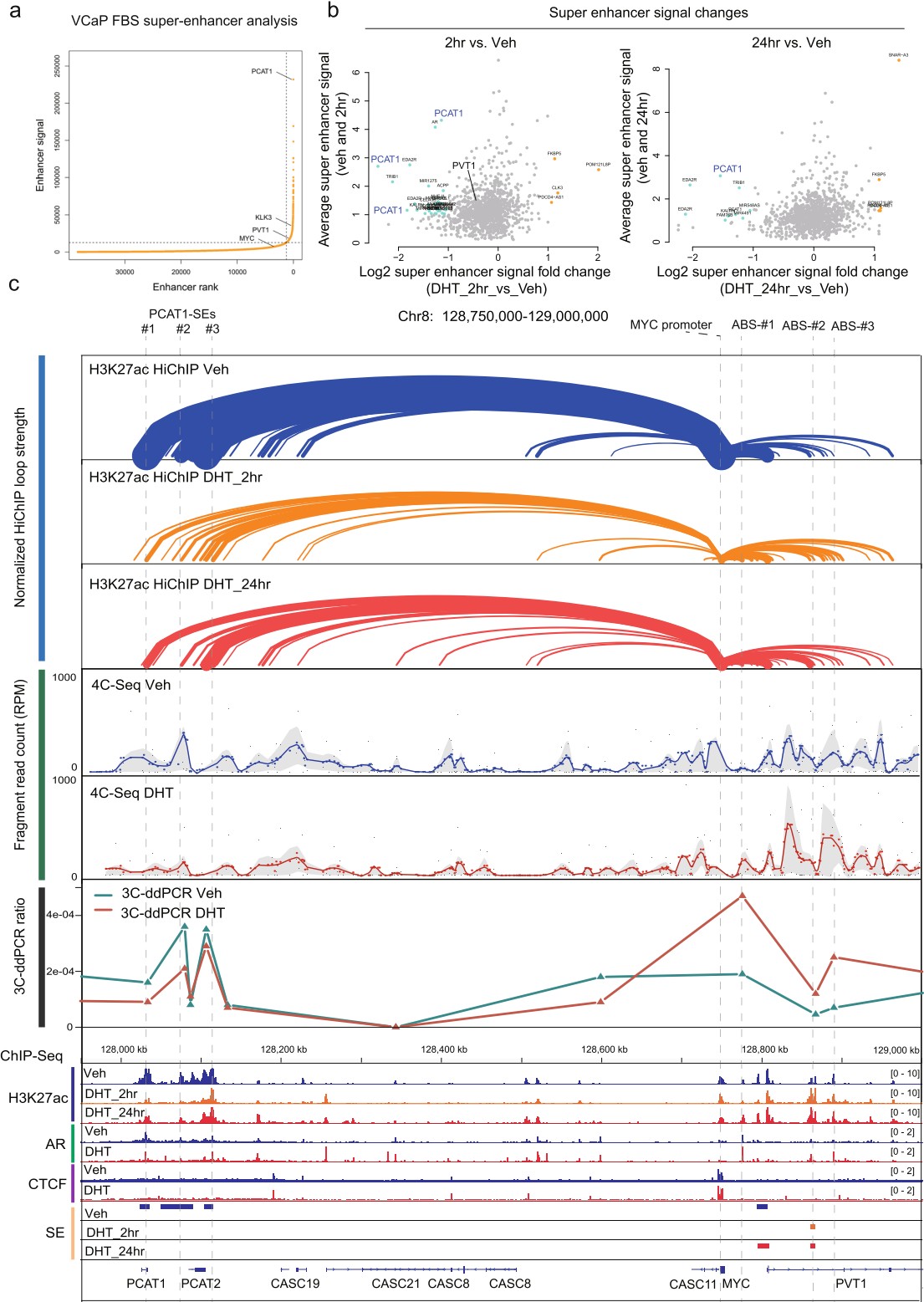

**Fig. 6 8q24 super-enhancers drive MYC transcription by distal interaction that is perturbed by DHT treatment. a** SE analysis of H3K27ac ChIP-seq data in VCaP cells cultured in regular medium. **b** H3K27ac signal fold change of all 1244 SEs under vehicle condition versus 2 h or 24 DHT stimulation. (**c**) Combined H3K27ac HiChIP and H3K27ac, 4C-seq, 3C-ddPCR, AR, and CTCF ChIP-seq annotation showing confident interactions anchored at *MYC* promoter. For H3K27ac interactions, loops with less than 10 mated reads were filtered out. The 4C-seq and HiChIP results showed DHT stimulation attenuated the looping from *MYC* promoter to the left arm (*PCAT1*-SEs (#1, #2, and #3)) but enhanced the looping from *MYC* promoter to the right arm (ABS-#1, ABS-#2, and ABS-#3). For 3C-ddPCR, the *MYC*-promoter was used as bait (constant) that is paired with primers specifically located in target regions.

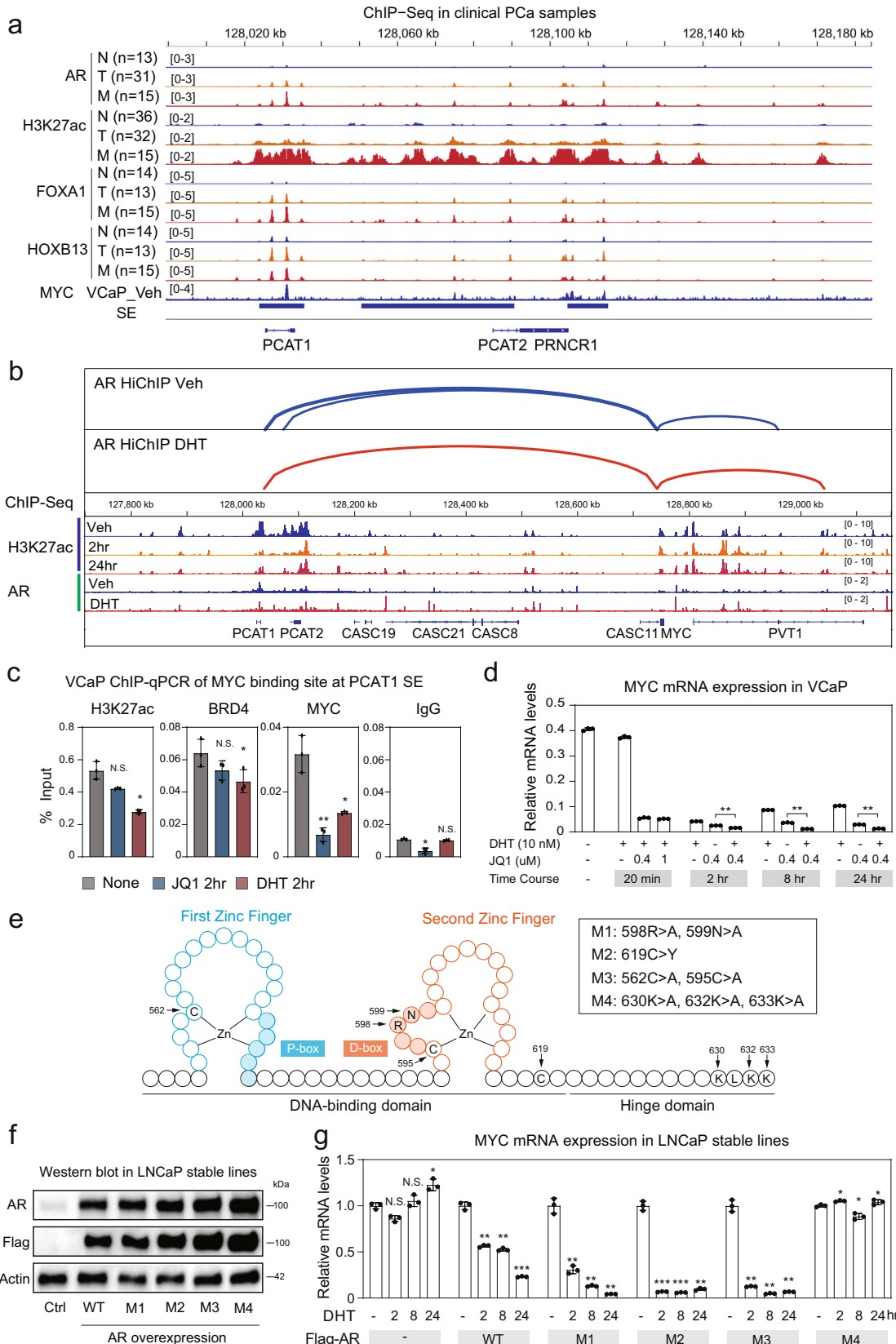

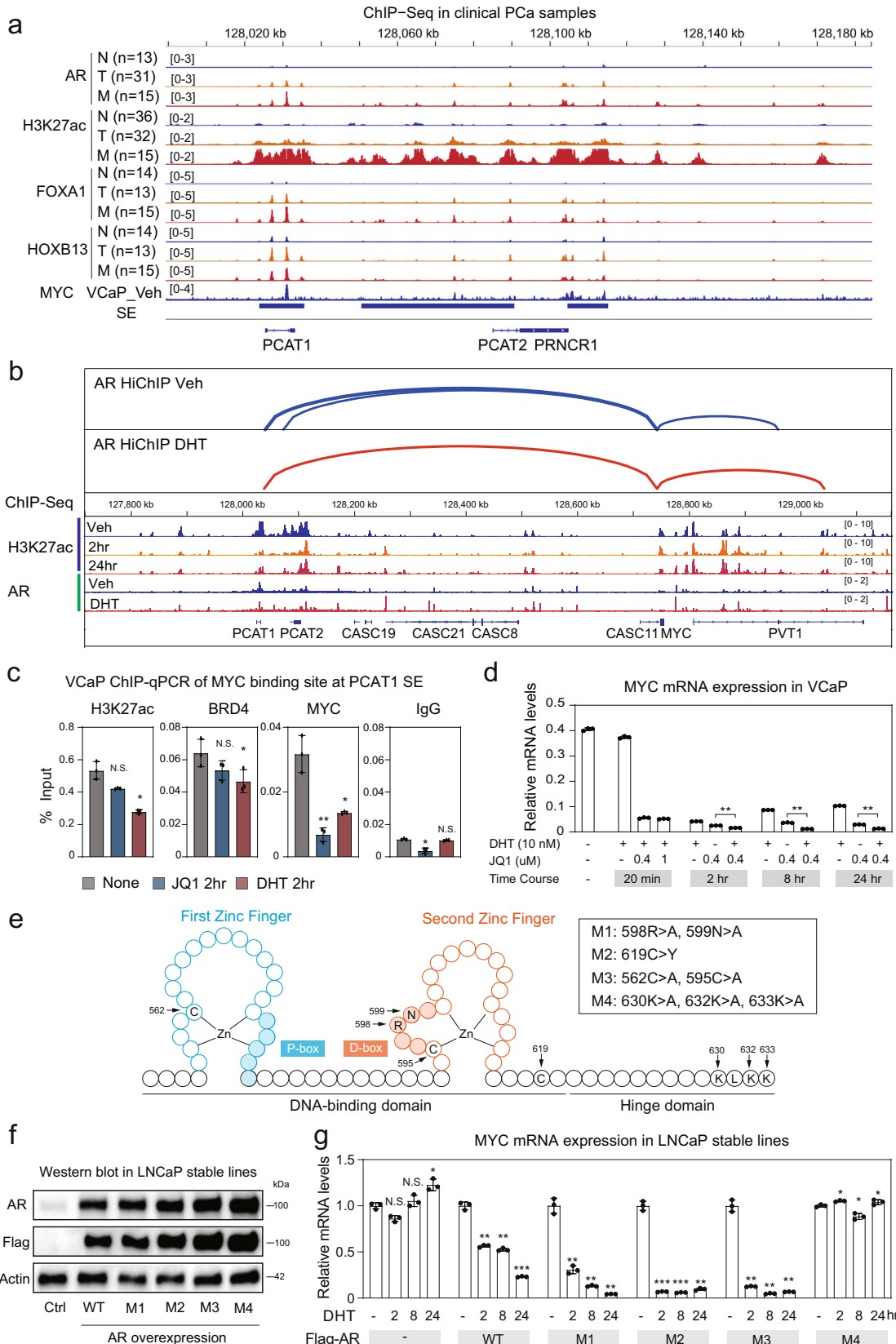

cofactors (Fig. S10b). In contrast, the M4 mutant, which is defective in nuclear localization, did not repress TMPRSS2 expression. Interestingly, the M4 mutant (as well as over-expression of the WT AR) decreased DHT-stimulated expression of PSA, but to a lesser extent than the other mutants, possibly

reflecting unique features and requirements for maximal activity of the PSA enhancer.

As expected, MYC became DHT-repressed in the WT AR overexpressing LNCaP cells (Fig. 7g). Significantly, MYC was even more markedly repressed by the M1, M2, and M3 mutants,

**Fig. 7 Convergence of dual AR regulatory functions on PCAT1 SEs and MYC regulation. a** Annotation of AR and co-factor binding at PCAT1-SE in clinical PCa. Binding profiles annotated: AR, H3K27ac, FOXA1, and HOXB13 (GSE130408). For each the profiling was based on average of normalized signals across the cases as indicated. Bottom track is an alignment of VCaP MYC ChIP-seq generated in this study. N: normal; T: primary PCa; M: metastatic CRPC. **b** Combined AR HiChIP, H3K27ac ChIP-seq and AR ChIP-seq annotation showing confident AR-mediated interactions anchored at *MYC* promoter. **c** ChIP-qPCR of H3K27ac, BRD4, and MYC at MYC binding site in *PCAT1* SE with BRD4 antagonist JQ1 (500 nM) or DHT (10 nM) treatment for 2 h in VCaP cells in androgen-depleted medium. **d** VCaP cells in androgen-depleted medium treated with DHT and/or JQ1. Total RNA was subjected to qRT-PCR analysis of MYC with GAPDH as internal control. **e**, **f** LNCaP-AR stable pools were generated to overexpress Flag-AR versus mutants: M1 (R598A-N599A: mutation at the DBD zinc finger 2 (ZF2) D-box that mediates AR dimerization); M2 (C619Y: mutation at AR DBD between ZF2 and hinge domain, defective in DNA binding but not nuclear localization; M3 (C562A-C595A: mutations in AR DBD at two zinc finger cysteine residues that are in ZF1 and ZF2, respectively); and M4 (K630A-K632A-K633A: mutation in the NLS (nuclear localization signal) at the hinge region, defective in both nuclear localization and transactivation. Ctrl: parental LNCaP. The Western blotting experiment was repeated independently three times with similar results. **g** Parental LNCaP (ctrl, −), LNCaP-AR WT, and mutants stable pools were cultured in androgen-depleted CDS medium for 2 days, and then subjected to androgen treatment (10 nM of DHT) as indicated. Total RNA was submitted to RT-PCR analysis of MYC (GAPDH as internal control) and the ratio was normalized to the vehicle-treated control, which was set at 1. For **c**, **d**, and **g**, Data are presented as mean values ± SD. For each test, $n = 3$ biologically independent samples. $P$ values were determined by two-sided Student's $t$ test. N.S., not significant; *, $P < 0.05$; **, $P < 0.01$; ***, $P < 0.001$. Source data are provided in Histogram and Immunoblot Source Data files.

but was not repressed by the M4 mutant. Together, these findings indicate that DHT-mediated repression of MYC activity is dependent on AR nuclear localization, but not binding to specific AREs, and support the conclusion that the MYC repression is due to redistribution and sequestration of transcriptional cofactors. Interestingly, the more potent MYC-repressive effect of the mutants that are defective in DNA binding relative to the WT-AR further indicates that AR binding at the *PCAT1* SE may be stimulatory, but not potent enough to fully compensate for the sequestration of cofactors by overexpressed AR. Finally, to confirm that loss of DNA binding did not prevent cofactor binding, we assessed WT and M2 mutant AR binding to BRD4 and MED1. Flag immunoprecipitation from LNCaP cells expressing the Flag-tagged WT and M2 mutant ARs brought down comparable amounts of BRD4 and MED1 (Fig. S10c).

One implication of these findings is that levels of coactivators including BRD4 may be limited in cells with high-level AR expression. Indeed, we found that LNCaP-AR cells were hypersensitive to JQ1 relative to parental LNCaP cells (Fig. S10d). We also addressed whether decreased MYC may contribute to the biphasic effects of androgen on proliferation, with high androgen levels being growth-suppressive. Consistent with previous data, DHT can stimulate proliferation of LNCaP cells cultured in medium with steroid-depleted FBS (charcoal-dextran stripped FBS, CDS medium), which is associated with an increase in MYC (Fig. S10e). In contrast, addition of further DHT to LNCaP cells in complete FBS medium can suppress proliferation, with modest MYC attenuation. These effects are more dramatic in LNCaP cells overexpressing AR (LA cells), where DHT markedly decreases MYC and cell proliferation (Fig. S10f). Significantly, over-expression of MYC in these cells (LAM cells) increases proliferation and blunts the effects of DHT, further supporting decreased MYC as a mechanism contributing to the biphasic effects of androgen on cell proliferation, and to the therapeutic effects of supraphysiological androgen therapy in CRPC.

## Discussion

The transcriptional activation functions of AR have been well-studied, but the mechanisms through which it acts to repress transcription, and the physiological significance of this repression, remain to be established. Through integration of RNA-seq and ChIP-seq data, we found that AR binding sites were not significantly associated with DHT-repressed genes. While this does not rule out direct AR binding mechanisms on a subset of genes, or possibly repression through low affinity or very distal sites, it supports coactivator redistribution as a major mechanism. We further found that MYC was the most rapidly and dramatically

repressed TF in response to androgen, that the MYC binding motif was highly enriched amongst DHT-repressed genes, and by BETA found that MYC binding was strongly associated with DHT-repressed genes. These findings indicate that MYC down-regulation is a further significant contributor to androgen-mediated transcriptional repression. The MYC gene in PCa is regulated by a prostate-specific SE overlapping the *PCAT1* gene, and we show that androgen represses MYC expression by disrupting the interaction between this SE and the *MYC* promoter. Significantly, AR binds to several sites in this SE, and this binding may contribute to the activation of this SE (see below). However, we found that ARs with mutations that abrogate DNA binding could still repress MYC expression, supporting the conclusion that redistribution of coactivators including BRD4 is driving the MYC repression (Fig. S11a).

In support of the physiological significance of this MYC repression, we found DHT treatment of CRPC xenografts could rapidly suppress MYC, a response that may contribute to the efficacy of supraphysiological androgen therapy in CRPC (Fig. S11b)[40,41]. Conversely, androgen deprivation in vitro and castration in vivo led to rapid and persistent increases in MYC. The expression of MYC and of MYC-regulated genes is also increased in CRPC, including in clinical samples without *MYC* gene amplification. This indicates that decreased AR activity in CRPC can be compensated by increased MYC, and that this may contribute to progression to CRPC after androgen deprivation therapy (Fig. S11b).

Previous data indicate that androgen can repress MYC in normal prostate epithelium, which may be part of an AR-driven program to drive terminal differentiation[42]. This repression may be through a distinct mechanism as the *PCAT1* linked *MYC* SE appears to be inactive in normal prostate epithelium, but it is possible that it is driving MYC at an earlier developmental stage that coincides with AR induction. Of note, activity of this enhancer appears to be AR-dependent, as there is prominent AR binding at several sites on this *MYC* enhancer, and it is inactive in AR negative PCa lines. Moreover, we also found AR binding to these sites and AR-mediated looping to the *MYC* promoter by AR HiChIP under basal conditions in androgen depleted medium, and a previous study found that RNAi mediated depletion of AR decreased MYC expression[11]. Together these findings indicate that AR may play a critical role in activation of this enhancer, and in subsequently fine-tuning its activity based on AR and androgen levels. Finally, AR can also bind to β-catenin, and previous studies indicate it can thereby decrease available nuclear β-catenin and suppress Wnt/β-catenin signaling, providing a further mechanism for fine-tuning MYC expression[43].

While MYC binding was associated with DHT-repressed genes, it was also associated with DHT-stimulated genes. Consistent with this observation, we found that ~25% of MYC binding sites detected by ChIP-seq overlapped with AR binding sites, indicating that reduced MYC at these sites may be compensated by increased AR binding and activity. This ~25% overlap is very close to the ~30% overlap observed in a previous study in LNCaP PCa cells, although this previous study found a larger fraction of AR binding sites overlapping with MYC sites than in our study (~25% versus ~7%)[19]. Remarkably, we found that these overlapping AR/MYC sites had markedly higher levels of H3K27ac than AR unique sites, indicating they are highly enriched for active promoters and enhancers. It is not clear whether AR and MYC are directly interacting at these sites, with the alternative hypothesis being that binding of AR or MYC is indirectly facilitated by MYC or AR, respectively. As H3K27ac levels are also high at MYC unique sites, we would favor the hypothesis that MYC binding primes these sites for subsequent AR binding, or alternatively that MYC binding is a mark for sites that have been primed by other TFs. AR binding may then further fine-tune activity at these sites. Of note, MYC overexpression in LNCaP PCa cells was found to induce partial reprogramming of the AR cistrome in a previous study, but the altered sites were primarily low affinity and binding at the majority of sites was not altered[19]. However, the MYC overexpression in this previous study did not substantially alter MYC binding to chromatin, making it difficult to interpret the results.

As expected, we found that MYC depletion by siRNA decreased the expression of androgen-repressed genes, consistent with a subset of these genes being MYC regulated. However, we found that MYC depletion also increased the expression of androgen-stimulated genes, and particularly androgen-stimulated genes lacking a MYC binding site. Conversely, MYC overexpression markedly suppressed the expression of androgen-stimulated genes. A previous study similarly found that MYC overexpression suppressed AR transcriptional activity[19]. One basis for this effect, similarly to the effect of androgen on MYC expression, may be coactivator redistribution. However, given the large number of genes modulated by MYC, an alternative mechanism is certainly MYC induction of one or a series of genes that more directly suppress AR activity.

As noted above, one clinical implication of these findings is that androgen deprivation therapies for PCa result in a compensatory increase in MYC, which may be important for initial tumor cell survival. Therefore, agents that can suppress MYC expression or activity may be most effective when used early in combination with androgen deprivation. Such agents may include a new generation of BET protein inhibitors and MYC-targeting compounds. Moreover, given the specificity of the *PCAT1* overlapping MYC SE for PCa cells, a further understanding of how it is regulated may yield strategies to selectively suppress MYC expression in PCa.

## Methods

**Cell line generation**. LNCaP AR overexpression stable line (LA) was from Dr. Matthew L Freedman (DFCI, Boston, MA 02215, USA), which was generated based on lentiviral infection of LV_AR_orf vector that expresses Flag-tagged AR and hygromycin B selection[2]. LNCaP AR and MYC overexpression stable line (LAM) was generated by lentiviral infection of LA cell line with the pCDH-MYC vector (Addgene, Plasmid #46970) using the Lenti-Pac HIV Expression Packaging system (Genecopiea, LT001), followed by hygromycin B and puromycin selection. LNCaP stable lines overexpression of AR WT and mutants were generated using AR mutants that were constructed based on DNA mutagenesis (QuikChange Lightning site-directed mutagenesis, Fisher, NC9620881), followed by lentiviral infection and hygromycin B selection.

**Antibodies and siRNA**. Antibodies used in these studies are as follows: MYC (rabbit polyclonal Santa Cruz sc-764; recombinant rabbit mAb clone Y69 Abcam

Ab32072), AR (rabbit polyclonal Santa Cruz sc-816; rabbit polyclonal Abcam Ab74272), β-tubulin (mouse mAb clone KMX-1 EMDMillipore MAB3408), PSA (rabbit polyclonal BioDesign K92110R), Vinculin (mouse mAb clone hVIN-1 Sigma V9264), H3K27Ac (rabbit polyclonal Abcam ab4729, rabbit polyclonal DIAGENODE Cat. C15410196), BRD4 (rabbit polyclonal Bethyl A301–985A), MED1 (rabbit polyclonal Bethyl, A300–793A), Flag (mouse mAb clone M2, Sigma Aldrich F3165), β-actin (mouse mAb clone AC-15 Abcam Ab6276). For immunoblotting, antibodies were used at 1:000 dilution. For IHC antibodies were used at 1:2500. For ChIP, antibodies were used at 4 μg per sample, or 5 μg per sample for HiChIP. MYC siRNA-1 was from Dharmacon (LU-003282-02-0002ON-TAR-GETplus Human MYC (4609) siRNA (#26). MYC siRNA-2 targeting sequence GCTTGTACCTGCAGGATCT was from Dharmacon (#1094). DHT was used at final concentration of 10 nM unless specified otherwise.

**Chromosome Conformation Capture (3C)**. 3C was performed as previously described[2]. Briefly, 4 × 15 cm dishes (two dishes for each assay) of VCaP cells in 5% CDS medium were treated with vehicle or DHT (10 nM, 2 h) and then fixed with 1% formaldehyde for 10 min, followed by quenching the reaction by glycine. Cells were lysed and pelleted nuclei were resuspended in 100 μl of 0.5% sodium dodecyl sulfate (SDS) and incubated at 62 °C for 10 min, followed by dilution in Triton X-100 to quench the SDS. Chromatin was then digested overnight at 37 °C with 500 U of PstI (NEB, R0140S), followed by 80 °C for 20 min. Each sample then underwent ligation in 1 ml with 8000 U of T4 DNA Ligase (NEB, M0202L) at room temperature. Proteinase K (30 μl of 20 mg/ml) (NEB, P8107S) was then added followed by incubation overnight at 65 °C. RNase A (15 μl) (10 mg/ml, Thermo-Fisher, EN0531) was then added followed by incubation for 45 min at 37 °C. Samples were then purified by phenol:chloroform:isoamyl alcohol (25:24:1; ThermoFisher, 15593-031) extraction and then ethanol precipitated.

dsDNA was quantified using Qubit (Model 3.0, Life Technologies). For quantitative 3C droplet digital PCR (ddPCR) analyses, 400 ng of VCaP-PstI 3C library DNA was used as template. The MYC-promoter was used as bait (constant) that was paired with primer specifically located in target regions. Primer sequences are in Supplementary Table S1. Specific probe signal was normalized to that of the copy number reference RPP30 that does not contain a PstI site in the amplicon. The ddPCR was performed on the Bio-Rad QX200 AutoDG Droplet Digital PCR System.

**Circular chromosome conformation capture (4C)**. Each 3C library DNA sample was digested with Dpn II (NEB, R0543S). The 4C ligation was in 10 ml with 50 μl T4 DNA ligase (400 U/μl, NEB) at room temperature overnight. DNA was then ethanol precipitated and purified with the QIAquick PCR purification kit (Qiagen, 28106). The 4C-DpnII library was then amplified on the Veriti 96-Well Thermo Cycler (Applied Biosystems). Limited PCR (round-1) was conducted with the MYC-promoter region as the bait. The PCR products were purified using PCR purification kit and re-amplified by limited PCR (round-2) with primers containing Illumina sequencing adapters (Read 1 and Read 2 adapters, respectively). Primer sequences are in Supplementary Table S1. 4C library purification was performed using Ampure beads (Fisher, NC9959336) and then analyzed by deep sequencing (85 M PE150 reads per sample) using the Illumina platform (Genewiz Inc.).

**Chromatin immunoprecipitation (ChIP) assay**. Cells were crosslinked in 1% formaldehyde for 10 min and quenched by glycine. Cell nuclear fraction was extracted by LB1 buffer (50 mM Hepes–KOH, pH 7.5; 1 mM EDTA; 140 mM NaCl; 10% glycerol; Igepal CA-630; 0.25% Triton X-100) followed by washing in LB2 (10 mM Tris–HCL,pH8.0; 200 mM NaCl; 1 mM EDTA; 0.5 mM EGTA) and LB3 buffer (10 mM Tris–HCl, pH 8; 100 mM NaCl; 1 mM EDTA; 0.5 mM EGTA; 0.1% deoxycholate; 0.5% N-lauroylsarcosine). Chromatin was sheared to 300–500 bp by Diagenode bioruptor sonicator and then incubated with antibody-conjugated protein A and G beads overnight at 4 °C with rotation. The antibodies for ChIP-seq were anti-MYC (Santa Cruz, sc-764) or anti-H3K27ac (Abcam, ab4729). Antibodies for ChIP-qPCR were anti-MYC (Abcam Ab32072), anti-AR (Abcam Ab74272), or anti-H3K27ac (DIAGENODE C15410196). All antibodies were used at 4 μg per sample. After washing in RIPA buffer (50 mM Tris, pH 7.6, 150 mM NaCl, 1 mM EDTA, 0.1% SDS, 1% Igepal CA630, 0.5% deoxycholate, protease inhibitors and RNase inhibitor), the beads were eluted in elution buffer (0.1 M NaHCO₃, 1% SDS and Proteinase K) for 8–16 h at 65 °C. ChIPed DNA was purified by phenol–chloroform extraction and then used for ChIP-qPCR or ChIP-seq library preparation.

**HiChIP**. HiChIP was performed as previously described[44]. Briefly, ~10 million cells were crosslinked in 1% formaldehyde for 10 min at room temperature and then quenched by 125 mM glycine. After washing in PBS, the crosslinked cells were lysed in Hi-C lysis buffer (10 mM NaCl, 10 mM Tris-HCl, 0.2% NP-40, and 1 × protease inhibitor) and digested by MboI restriction enzyme (NEB-R0147). The biotin-dATP (Thermo 19524016) was incorporated to DNA in a fill-in master mix containing DNA polymerase I (NEB-M0210). After ligation and sonication, the sheared chromatin was incubated together with Protein A beads and anti-H3K27ac antibody (Abcam, ab4729, 5 μg per sample) or AR antibody (Santa Cruz sc-816, 5 μg per sample) for overnight at 4 °C with rotation. ChIPed DNA was purified from

the eluted chromatin using Zymo DNA Clean & Concentrator kit. Biotin-labeled DNA was captured by Streptavidin C-1 bead (Thermo Fisher, Cat#65002) and fragmented by Tn5 enzyme to generate libraries for high throughput sequencing. Next-generation sequencing was conducted on a HiSeq4000 machine with 150 bp paired-end reads.

**TaqMan real-time RT-PCR**. RNA isolation was carried out using the TriZOL reagent (Ambion) and the qRT-PCR analysis on gene expression was performed on the TaqMan StepOnePlus Real-Time PCR Master system (Applied Biosystems). The following TaqMan primer-probe sets were purchased as inventoried mixes from Applied Biosystems: MYC (Hs00153408_m1), AR (Hs00171172_m1), PSA/KLK3 (Hs02576345_m1), TMPRSS2 (Hs01120965_m1), NKX3-1 (Hs00171834_m1), LMNB1 (Hs01059210_m1), RAD51AP1 (Hs01548891_m1), PVT1 (Hs01069041_m1), PCAT1 (Hs04275836_s1), CASC8 (Hs03666772_g1), FAM84B (Hs00326521_m1), LINC00977 (Hs01596349_m1), GSDMC (Hs00937071_m1). The CCAT1 gene TaqMan primer set is forward primer: GGCCAGCCCTGCCACT; reverse primer: CAGTTTTCAAGGGATTTTAGGAGAA; and probe: ACCAGGTTGGCTCTGTA TGGCTAAGCGT. The inventoried internal control was GAPDH (VIC-TAMRA labeled, Life Technologies, 4310884E).

**Cell culture, transfection, and proliferation assay**. LNCaP was grown in RPMI-1640 containing 10% FBS (GIBCO, 10437-028). LNCaP cell lines overexpressing AR (LNCaP-AR, or LA) were maintained in medium containing hygromycin B (200 μg/ml). LNCaP cell lines overexpressing AR and MYC (LNCaP-AR-MYC, or LAM) were maintained in medium containing hygromycin B and puromycin. VCaP cells were grown in Dulbecco's modified Eagle's medium (DMEM) medium containing 10% FBS. For androgen-starving conditions, cells were grown in medium containing 5% CDS (charcoal-dextran stripped FBS). Plasmids and siRNA transfections were carried out using the Lipofectamine 2000 (Invitrogen), following the manufacturer's directions. Cell proliferation was performed using the CellTIter-Glo Luminescent cell Viability Assay Kit (Promega, G7572).

**VCaP xenograft**. ICR/scid male mice (6–8-week-old, IcrTac:ICR-Prkdc<scid>, from Taconic Biosciences, Inc., 273 Hover Avenue, Germantown, NY 12526, USA) were used to generate xenografts. VCaP xenografts were established in the flanks of male SCID mice by injecting ~2 million cells in 50% Matrigel. When the tumors reached ~1 cm, biopsies were obtained and then the mice were castrated. Groups for testosterone test are: control castrated VCaP xenografts (C); castrated VCaP xenografts with testosterone (5 mg/mouse/day for 3 days) (T); castrated VCaP xenografts with testosterone and enzalutamide (testosterone 5 mg/day for 3 days, and then enzalutamide 40 mg/kg/day for 3 days) (T + E). For IHC, additional biopsies were obtained 4 days after castration, and the tumors were harvested at relapse. Frozen sections were examined to confirm that the samples used for RNA extraction contained predominantly nonnecrotic tumor. Fixed sections were subjected to IHC staining for MYC (Santa Cruz sc-764) or PSA (BioDesign K92110R), both at 1:2,500 dilution. All animal experiments were approved by the Beth Israel Deaconess Institutional Animal Care and Use Committee and were performed in accordance with institutional and national guidelines. Tumor-bearing mice were sacrificed before the tumors reach the size considered to be distressing (2 cm). Animal stress and discomfort were minimized by the use of methods recommended by the AVMA Panel on Euthanasia and approved by the Institutional Animal Care and Use Committee. Mice were monitored daily by animal facility staff for tumor size and overall health. When necessary, euthanasia was carried out by personnel in the animal facility by CO₂ asphyxiation using an AVMA approved method and apparatus.

**RNA-seq analysis**. Total RNA was extracted from VCaP cells using TriZol reagent and whole transcriptome sequencing was conducted on Novaseq6000 S4 flowcell for PE150 sequencing (Novogene Corporation Inc., Sacramento, CA 95817). Raw sequencing reads were first trimmed by Trim Galore (https://github.com/FelixKrueger/TrimGalore) with parameters "-q 20–phred33 –stringency 4 –length 20 -e 0.1". The clean reads were then aligned to hg19 human genome using STAR (version 2.4.2a)[45] with parameters "–outSAMattributes NH HI NM MD –outSAMstrandField intronMotif –quantMode GeneCounts". Reads per kilobase per million mapped reads (RPKM) were then calculated based on gene read count and GENCODE v24 GRCh37 annotation. R package clusterProfiler[46] was used to perform KEGG/GO enrichment analysis and Gene Set Enrichment Analysis (GSEA). To identify the differentially expressed genes in Fig. 1c, d, R package limma and DESeq2 were used for microarray data and RNA-seq data respectively.

**ChIP-seq analysis**. To generate ChIP-seq library, 5 ng ChIPed DNA was processed with the Rubicon ThruPLEX-FD kit and sequenced as 75 bp single-end reads. ChIP-seq reads were first trimmed by Trim Galore in the same option setting with RNA-seq and then aligned to hg19 human genome by Bowtie2 (version 2.2.1)[47] with default parameters. The bam files were then subjected to MACS2[48] for peak calling with the parameter "–SPMR" on and "–keep-dup = 1". Significant peaks were identified by q-value < 0.05. UCSC bedGraphToBigWig tool was used to convert Resultant bedgraph files to bigWig files, which were then used for peak shape visualization in IGV (version 2.8.2) and heatmap analysis by deeptools

(version 3.3.1)[49]. Cistrome SeqPos[50] and R package PWMEnrich were used to identify enriched transcription factor binding motifs in a set of peaks. Binding and Expression Target Analysis (BETA) was used to assess the regulatory capacity of the given ChIP-seq peaks on gene expression[51]. Super-enhancers were recognized from H3K27ac ChIP-seq peaks by Ranking Of Super Enhancer[52].

**Whole-genome bisulfite sequencing analysis**. Two PCa tissues whole-genome bisulfite sequencing (WGBS) data sets were used in this study. The paired-end WGBS data set from the prostate tissue of four healthy donors and five prostate cancer patients were obtained from the authors[37]. Samples of tissue adjacent to the tumor from four of the prostate cancer patients were also included. The data processing was conducted as previously described. Briefly, after trimming by Trim Galore (version 0.4.4_dev), trimmed reads were mapped to the hg19 genome reference using Bowtie2 and Bismark v0.19.0[53]. Another 208 pairs of primary PCa and adjacent tissue WGBS data were from the Genome Sequence Archive for Human under the accession number PRJCA001124[38]. Output files of Bismark were downloaded for downstream analysis. R package "dmrseq" was used to process the count data and plot average methylation signal of super-enhancer regions[54].

**4C-seq data analysis**. The PE reads were first aligned to hg19 human reference genome using Bowtie2 (version 2.2.1). The resultant bam files were then used for downstream analyses by R package "Basic4Cseq"[55]. A fragment library was created by the function "createVirtualFragmentLibrary" using the parameter first-Cutter = "ctgcag", secondCutter = "gtac". 4C bam files were then loaded to initiate a Data4Cseq object and the viewpoint was set at MYC promoter region. The raw read count was RPM-normalized for the comparability between different samples. A near-cis plot was finally generated to cover the MYC promoter and PCAT1 super-enhancer regions.

**HiChIP data analysis**. Pair-end HiChIP reads were first trimmed by Trim Galore and then aligned to hg19 human reference genome by HiCUP pipeline[56]. The default settings (Quiet:0; Keep:0; Zip:1; Longest: 700; Shortest: 50) were used to filter experimental Hi-C artefacts and other uninformative di-tags. The resultant bam file was used to generate valid pairs by command "samtools sort -n hicup.bam | bamToBed -i -bedpe | awk 'OFS = "\t" {print $7,$1,int(($2 + $3)/2), $9, $4, int(($5 + $6)/2), $10}' > HiChIP.allValidPairs" for downstream loop calling. HiChIP loops were identified by hichipper pipeline[57] using valid HiChIP read pairs and pre-determined ChIP-seq peaks. The corresponding ChIP-seq peaks were used to locate HiChIP loop anchors during loop calling.

**TCGA PCa and Quigley CRPC RNA-seq data analysis**. TCGA PCa RNA-seq data (https://portal.gdc.cancer.gov/) were processed by STAR2/RSEM pipeline to generate TPM values for each sample. TPM values of CRPC RNA-seq data were downloaded from http://davidquigley.com/prostate.html [1]. Before gene expression level comparison, the AR or MYC amplified samples were removed. MYC-upregulated gene z-score of each participant was defined as the sum of z-scores of 136 MYC-upregulated genes. For each gene, $z\text{-}score = (x − \mu)/\sigma$; $x$ indicates pre-normalized gene expression level, $\mu$ indicates study mean of gene expression and $\sigma$ indicates study standard deviation of gene expression.

**Analysis of ChIP-seq and ATAC-seq data in clinical PCa and adjacent tissues**. The 268 ChIP-seq or ATAC-seq data of clinical PCa and adjacent tissues were downloaded from GEO under accession number GSE130408. ChIP-seq or ATAC-seq data were aligned to hg19 human reference genome by Bowtie2 (version 2.2.1). The bam files were then subjected to MACS2 for peak calling with the parameter "–SPMR" on and–keep-dup = 1. UCSC bedGraphToBigWig tool was used to convert Resultant bedgraph files to bigWig files. The bigWig files of each group were averaged and merged to a single bigWig file by a custom shell script for further visualization.

**Statistical analyses**. Data in bar graphs represent mean ± SD of three technical replicates and are representative of at least three independent experiments. Results for immunoblotting are representative of at least three independent experiments.

**Reporting summary**. Further information on research design is available in the Nature Research Reporting Summary linked to this article.

# Data availability

The data sets analyzed in this study are summarized in Supplementary Table S2. The high throughput sequencing data generated in this paper have been deposited to GEO under accession number GSE157107 and are now available. The remaining data sets analyzed in this study are public domain, and are listed here. AR, FOXA, HOXB13 and H3K27ac ChIP-Seq data of PCa tissues are from GSE130408. TCGA-PRAD RNA-Seq data from Genomic Data Commons Data Portal (https://portal.gdc.cancer.gov/). Quigley CRPC RNA-seq data is available at http://davidquigley.com/prostate.html. RNA-Seq of VCaP cells with AR knockdown is from GSE82223. VCaP xenograft RNA-Seq data is from GSE56829. LNCaP RNA-Seq data is from GSE125014 and GSE114267. LNCaP Hi-C data is from GSE105557.

ATAC-Seq data of TCGA pan-cancer tissues are available at https://gdc.cancer.gov/about-data/publications/ATACseq-AWG. H3K27ac ChIP-Seq in cancer cell lines are downloaded from ENCODE (https://www.encodeproject.org/chip-seq-matrix/?type=Experiment&replicates.library.biosample.donor.organism.scientific_name=Homo%20sapiens&assay_title=Histone%20ChIP-seq&assay_title=Mint-ChIP-seq&status=released). Whole-genome bisulfite sequencing data of PCa tissues are from https://ngdc.cncb.ac.cn/gsa-human/browse/HRA000099. VCaP GRO-Seq data is from https://www.ncbi.nlm.nih.gov/geo/query/acc.cgi?acc=GSE84432. AR and BRD4 ChIP-Seq data of VCaP cells are from GSE55062. AR ChIP-Seq data of LNCaP cells is from GSE83860. The remaining data are available within the Article, Supplementary Information or Source Data file. Source data are provided with this paper.

## Code availability

All the software used in this study is published and cited in the Methods section. The custom scripts used in this study are available at https://github.com/Haiyangg/Script_for_MYC_project[58]. The corresponding DOI is as follows: https://doi.org/10.5281/zenodo.5638091.

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

## Acknowledgements

This work was supported by grants from NIH (K99/R00 CA135592) and DOD (W81XWH-14-1-0016) to S.Y.C., and NIH P01 (CA163227) and SPORE in Prostate Cancer P50 grant (CA090381) to S.P.B. It was also supported by NSERC discovery grant (498706 to H.H.H.), CIHR operating grants (142246, 152863, 152864 and 159567 to H.H.H.), Terry Fox New Frontiers Program Project Grant (1090 P3 to H.H.H.). H.H.H. holds Joey and Toby Tanenbaum Brazilian Ball Chair in Prostate Cancer Research. H.G. is supported by Taishan Scholar Program of Shandong Province (NO.tsqn201812136) and National Key Research and Development Project of China (2019YFA0111400). Y.M.W. is supported by an overseas research scholarship from the China Scholarship Council (CSC) led by the Ministry of Education of China. M.L.F. is supported by the Claudia Adams Barr Program for Innovative Cancer Research, the National Cancer Institute (R01CA251555, R01CA193910), the Department of Defense (W81XWH-19-1-0565), the H.L. Snyder Medical Research Foundation, the Donahue Family Fund, the Mayer Foundation, and the Cutler Family Fund for Prevention and Early Detection. We thank the help from Dr. BaiLin Wu (Institutes of Biomedical Sciences, Fudan University, Shanghai, 200433, China; Boston Children's Hospital and Harvard Medical School, Boston, MA 02115, USA), Drs. Xin Yuan and Jiaqian Liang (Hematology-Oncology Division, Department of Medicine, Beth Israel Deaconess Medical Center and Harvard Medical School, Boston, MA, 02215, USA), and Dr. Jianhua Luo (Department of Pathology and Surgery, University of Pittsburgh School of Medicine, Pittsburgh, PA 15261, USA). We also appreciate Arvinus for providing the compound (AR degrader ARCC-32).

## Author contributions

The authors made the following contributions: conceptualization, H.G., S.C., S.P.B.; methodology, H.G., H.H., S.C., S.P.B.; software, H.G., Y.W., H.H.; validation, H.G., Y.W., H.H., S.C., S.P.B.; formal analysis, H.G., Y.W., H.H., S.C., S.P.B.; investigation, H.G., Y.W., Z.W., L.W., S.A., M.N., H.H., S.C., S.P.B.; resources, S.S., J.W.R., A.S., M.P., K.K., J.S., M.F., H.H., S.C., S.P.B.; data curation, H.G., Y.W.; writing original draft, S.C.; review and editing, H.G., H.H., S.C., S.P.B.; visualization, H.G., H.H., S.C., S.P.B.; supervision, H.H., S.C., S.P.B.; project administration, H.G., H.H., S.C., S.P.B.; funding acquisition, H.G., H.H., S.C., S.P.B.

## Competing interests

The authors declare no competing interests.
