## [Peer Review File · Nature Communications]

Androgen Receptor and MYC Equilibration Centralizes on Developmental Super-EnhancerReviewers' Comments:

Reviewer #1:

Remarks to the Author:

The manuscript by Guo et al analyzed the crosstalk between AR and MYC. Authors use three human PCa cell lines expressing and overexpressing AR and MYC and analyzed MYC downregulation by DHT indicating that androgen deprivation therapy induces MYC expression. Authors use RNA-seq and ChIP-seq in combination with knockdown of MYC and AR. In addition, AR Hi-ChIP was used to analyze a super-enhancer with binding profiles of AR, H3K27ac, FOXA1, HOXB13 and MYC to detect changes of DHT-regulated TADs by time.

Mutants of AR indicate that the DHT mediated inhibition of MYC is dependent on the AR nuclear localization but not its DNA binding function. GSEA analyses indicate that MYC depletion is enriched for DHT-repressed genes and MYC-activated genes are repressed by DHT. In line with this ADT indicates an activation of the MYC pathway. AR antagonists confirm authors suggestions including data obtained from mouse xenografts.

In patient samples and PDX, MYC expression is increased in CRPC

Authors suggest that coactivators may be sequestered by AR leading to loss of MYC dependent gene activation. Thus, decreased AR activity in CRPC can be compensated by increased MYC expression, which may contribute to progression of CRPC under ADT.

Major points:

1. The hormone concentrations (DHT and R1881) must be added for each experiment. Unclear is whether supraphysiological levels were used.

Page 4. Define androgen-starved. Is charcoal treatment meant or FBS-free media?

2. Missing are the transcriptome and ChIP-seq analysis of the cell lines under androgen-starved condition. Are more AR binding sites occupied in AR high expressing cells? Please show the basal level data without addition of androgens.

3. Authors used samples of CRPC patients. Unclear is whether the patients underwent ADT prior the analysis. Authors should make a statement for this issue. Authors imply that since in CRPC MYC is overexpressed, MYC to be a key factor in CRPC.

4. Authors reveal that loss of MYC accounts for 8% of DHT repressed genes.

In this context authors should tune down their too generalized statements about DHT mediated-repression of gene expression is mediated by MYC. Also the statement on page 9 should be tuned down: "MYC may have a large impact on the AR transcriptome"

5. These gene set of 8 % should be analyzed in more detail to identify pathways.

6. Please show data whether change of BRD4 recruitment correlate to AR repressed genes and analyze pathways of this subset of genes in IPA or GSEA.

7. Authors focus on one pioneering factor FOXA1. Authors should analyze their ChIP-seq data genome-wide also for sequence binding motifs of the additional two known pioneering factors HOXB13 and GATA2.

8. Please show the knockdown of AR at protein level.

Minor points

Authors used knockdown and not knockout. Therefore, please replace "loss of MYC" by "reduced MYC"

Page 4: "Androgen stimulation" should, be replaced. E.g. androgen treatment or androgen-mediated stimulation.
Androgen itself cannot be stimulated.

Page 17: replace "DHT repression of MYC" by DHT-mediated repression of MYC activity

Authors state that MYC inhibits AR very generally driven by coactivator redistribution, although only one specific (BRD4) is meant. Please tune down some sentences generalizing coactivators.

Many sentences should be shortened to avoid misinterpretation.

Reviewer #2:

Remarks to the Author:

The manuscript by Guo H et al. elucidated the intricate interactions between AR and MYC in prostate cancer. They both compete for coactivators and function cooperatively to maintain a stable expression of genes regulating multiple cellular functions. Their findings showed that increased MYC in response to androgen deprivation contributes to the development of CRPC, while decreased MYC may contribute to tumor regression in response to supraphysiological androgen therapy in men with CRPC. The authors performed a wealth of bioinformatic analyses to support their findings, which are biologically interesting and clinically relevant. However, substantial effort should be made to improve the clarity of the manuscript by providing more technical details about various analyses and proofreading by a native speaker.

Major comments:

1. A lot of bioinformatics analyses were performed throughout the manuscript, but substantial clarifications should be made. The description of Methods should be more detailed. For example, the trimming steps in the sequencing data analysis, the software/package version information, as well as the detailed specifications of parameters/arguments should be provided. The statistic tests used should be provided and properly phrased to support the conclusions, e.g., Figure 1C, 1D, Figure 4B-E, G, H, J, K, Figure 5A-B, I, Figure 7C, D, and G.

2. Specifically in Figure 1:

1) Figure 1C-D: what is the method or tool for identifying Androgen activated and repressed genes using the VCaP RNA-seq data? How consistent are the differential/dysregulated genes identified by microarray and RNA-seq data in LN, VC and VA? Why different cutoffs on fold changes were used (1.5 and 2)? Also, the common cutoffs on the statistical significance for selecting differentially expressed genes should be $FDR < 0.05$, instead of $P \text{ value} < 0.05$.

2) Figure 1E: AR binding peaks were integrated with differential expression for the analysis, but there was no data showing the binding patterns of AR. Were the binding peaks identified using ChIP-seq data at 2hr, 10hr and 24hr, respectively? More information should be provided to illustrate/compare the genome-wide distributions of AR binding sites (promoters, gene body, intergenic, etc.) in LN, VC and LA cell lines.

3) Figure 1F: please explain what it means by "genes classified based on the distances to AR and H3K27ac (GSE96652) peaks". Also, the author showed the expression of DHT-repressed genes that are associated with H3K27ac sites that do not overlap AR sites (H3K27ac_only) decreased most significantly in response to DHT. Are there any DHT-repressed genes that are not associated with H3K27ac and AR sites?

3. In Figure 6B, the author showed that "The PCAT1 SE has three smaller sub-SEs, and H3K27ac at each site and overall was markedly decreased upon DHT stimulation". A statistic test should be provided to support the "markedly" conclusion since the figure only showed the log2 fold changes and

average signals of super-enhancers.

4. The authors should carefully check all the terms and abbreviations, and keep them consistent throughout the manuscript, such as "androgen-repressed genes", "DHT-repressed genes" and "DHT-downregulated genes", "LNCaP-AR" and "LA", "VCaP" and "VC", "ChIP-Seq" and "ChIP-seq". Some ambiguous terms may lead to misunderstanding, such as "AR activity" in "AR activity at MYC independent genes", "regulatory interaction" in "Locus-wide repression of the 8q24 TAD by androgen correlates with the decline in regulatory interaction".

Minor comments:

1. Figure 2A-B: how were the 1875 human TFs selected? A proper citation of the literature or database is needed.

2. In Figure 5G, there is no difference in the size of the dots denoting KEGG pathways with gene ratios of 0.05 and 0.10.

3. Typos, e.g.:

1) AR drives terminal differentiation. "dives" -> "drives"?

2) In the Method section "TCGA PCa and Quigley CRPC RNA-Seq data analysis", "TMP values" should be "TPM values".

3) "MYC up-regulated" should be "MYC upregulated" to be consistent with "DHT_downregulated".

Reviewer #3:

Remarks to the Author:

In this manuscript, the authors showed that DHT treatment repressed MYC expression and androgen deprivation increased MYC expression. They concluded that increased MYC in response to androgen deprivation contributes to castration-resistant prostate cancer, while decreased MYC may contribute to responses to supraphysiological androgen therapy. The novel finding such as super-enhancer relevant results in Figure 6 and Figure 7 is interesting. The authors could emphasize this major finding and add more mechanism results in the main Figures to enhance the novelty in this paper. However, most the phenomenon in the manuscript has been reported. For example, Lam et al (Eur Urol. 2020 Feb;77(2):144-155.) reported that supraphysiological testosterone treatment impaired cell cycle via robust downregulation of Myc-E2F pathway in ENZR PDXs. Monga et al (Sci Rep. 2020 Apr 20;10(1):6649) showed that c-MYC was upregulated in enzalutamide-resistant prostate cancer. Therefore, the authors should make effort to gain more novel mechanistic insight in the main Figures to specify the new findings. Overall, although the phenomenon and clinical significance reported in the manuscript are not very new, the equilibration of AR and MYC during anti-AR therapy and supraphysiological androgen therapy is interesting. The author could specify the finding and enhance the logic to present data in the main Figures.

1. In Figure 1C, the authors concluded that the number of androgen-activated/repressed genes in LNCaP cells is lower than in VCaP cells and AR-overexpressing LNCaP cells. However, there are some concerns for these data. Firstly, the microarray data and RNA-seq data of LNCaP, VCaP, and AR-overexpressing LNCaP cells were not acquired at the same conditions. The androgen concentration/treatment time, cell culture condition and microarray/RNA-seq platforms may influence the results which could affect the conclusion. For example, this manuscript shows that the number of androgen-activated/repressed genes in LNCaP cells is about 200, but Wang et al (Cell. 2009 Jul 23;138(2):245-56) reported that the number of DHT upregulated genes in LNCaP cells is about 400. If the authors chose this microarray data, the conclusion might be different. Therefore, the authors should be cautious with their conclusion.

2. Figure 1F and Figure 1G are confusing. In Figure 1F, the P value looks very significant in the middle panel, but the fold change is minimal. In Figure 1G, what is the value "Change = 0.0239". What does it mean? It is the fold change or p = value? I also didn't get the main point about Figure 1G. The authors should make it clearer and easier to understand.

3. In Figure 2D- F, could the authors comment on why DHT regulated MYC expression in VCaP cells but not in LNCaP cells? It is because AR level is very low in LNCaP cells? If so, AR level should be negatively correlated to MYC expression. Is this true in the patient's database such as TCGA?

4. The author showed that all MYC binding sites were decreased when treating with DHT (Figure 2J). However, in Figure 2M, why most of MYC up-regulated genes are not overlapped with DHT downregulated genes?

5. The statistical significance in the bar graphs such as Figure 1A, Figure 2D and others should be labeled.

Reviewer #4:

Remarks to the Author:

Androgen Receptor and MYC Equilibration Centralizes on Developmental Super-Enhancer

By Guo et al.

In their manuscript, Guo and colleagues provide a detailed analysis of the epigenetic mechanisms governing the androgen receptor-MYC signaling axis in prostate cancer. To this end they perform integrated analyses of newly generated ChIP-seq and RNA-seq data of cell line based prostate cancer models and employ publicly available datasets to support their observations. They thereby provide insights into the PCAT1 super-enhancer driven expression of MYC in prostate cancer, which is abrogated by AR activation leading to MYC repression and BRD4 redistribution. It is suggested that these processes thus indirectly promote the lower expression of genes suppressed by DHT treatment, which are a primary focus of the investigation. Overall, the in vitro experiments are well conducted, include relevant controls and appear technically sound. Comparable effects are observed in a cell line xenograft.

However, the novelty of the results is limited by the fact that several previous papers have investigated the connection between AR and MYC signaling in prostate cancer. Without being an expert in the prostate cancer field, it has been already reported in detail how MYC overexpression reprograms AR chromatin occupancy (Barfeld et al. EBioMedicine 2017). In addition, there is apparently some controversy in the field with respect to the role of MYC as there is no effect on AR as shown in a mouse model (Kim et al. Oncogene 2012) and in cell lines models AR knockdown has been shown to reduce the expression of MYC (Gao et al. Plos One 2013). Taking into account the body of published data the excitement for the presented data is limited. Also, while the data presented provide insight into the regulation of gene expression following DHT treatment and AR-MYC interplay several points remain to be further clarified, especially regarding the mechanism by which AR leads to MYC repression without DNA binding or dimerization capacity.

Major points:

1. It seems puzzling in Fig. 7F/G that AR is able to suppress MYC expression even in the absence of DNA binding or dimerization capacity (M1-3), but fails to do so if nuclear localization is mitigated (M4). Several additional experiments would help to elucidate the underlying biology:

- Transcriptomic data of LNCaP wt, M1-4 +/- DHT would be very helpful to assess the overall effect of the mutants on AR signaling and e.g. loss of induction of AR stimulated genes in M1-3 despite effects on MYC.
- Can it be shown that DNA binding/dimerization are indeed defective for M1-3 in the context

presented?

- Enzalutamide blocks binding of androgens to AR, binding of AR to DNA and nuclear translocation of AR. It would be helpful to see whether AR translocation is inhibited by enzalutamide in Fig. S2B/5A e.g. by IF. Do the same results with alternative AR inhibitors that do not prevent nuclear translocation (e.g. bicalutamide)?

2. It is suggested that MYC and AR compete for co-activators like BRD4 in the context of DHT treatment.

- Can overexpression of co-activator BRD4 mitigate the effects on target genes expression during DHT treatment?

- Fig. S10A suggests that AR not bound to DNA may still sequester co-activators, which seems in line with results from DNA binding defective variants (Fig. 7G). Can binding of non-DNA-bound AR to e.g. BRD4 be shown (e.g. co-IP or PLA +/- DHT)?

- Fig. 7C: Neither JQ1 nor DHT appear to strongly reduce BRD4 at the PCAT1 SE site and also lack a formal test of significance compared to vehicle. This effect appears much smaller than the effect observed on MYC expression. Please explain?

3. What is the translational relevance of these findings? This a topic that has not been touched too much although some of the findings might be of therapeutic relevance such as the increased binding of BRD4 at AR sites upon DHT treatment. This would be a major point to increase the impact of the study and to enhance its novelty.

Minor points:

1. Fig. 4A-E: Based on LA and LAM MYC expression decreases AR-only genes KLK3 and TMPRSS2. Investigation and side-by-side comparison with parental LNCap or ideally LNCap-MYCoE would be helpful to further judge the MYC AR interplay.

2. An overview/table of the data sets used indicating cell line/sample, treatment (compound, concentration, time), investigated marker, technology and source would be helpful.

3. The impact on cell proliferation of DHT/Enz has been shown for some of the cell lines used. To get a clearer picture about the impacts on global cellular phenotype viability assays assessing the impact of DHT concentrations and androgen depletion (CDS medium) on cell growth in VCAP, LNCaP, LA, LAM and LAM1-4 would be helpful.

4. If loss of MYC expression contributes to 7-8% of DHT suppressed genes, by what is the rest suppressed by?

5. Both amplification of AR and of MYC occur in CRPC. Are these independent events or do they co-occur?

REVIEWER COMMENTS

Reviewer #1, expert in AR signalling and prostate cancer (Remarks to the Author):

The manuscript by Guo et al analyzed the crosstalk between AR and MYC. Authors use three human PCA cell lines expressing and overexpressing AR and MYC and analyzed MYC downregulation by DHT indicating that androgen deprivation therapy induces MYC expression. Authors use RNA-seq and ChIP-seq in combination with knockdown of MYC and AR. In addition, AR Hi-ChIP was used to analyze a super-enhancer with binding profiles of AR, H3K27ac, FOXA1, HOXB13 and MYC to detect changes of DHT-regulated TADs by time.

Mutants of AR indicate that the DHT mediated inhibition of MYC is dependent on the AR nuclear localization but not its DNA binding function. GSEA analyses indicate that MYC depletion is enriched for DHT-repressed genes and MYC-activated genes are repressed by DHT. In line with this ADT indicates an activation of the MYC pathway. AR antagonists confirm authors suggestions including data obtained from mouse xenografts.

In patient samples and PDX, MYC expression is increased in CRPC

Authors suggest that coactivators may be sequestered by AR leading to loss of MYC dependent gene activation. Thus, decreased AR activity in CRPC can be compensated by increased MYC expression, which may contribute to progression of CRPC under ADT.

Major points:

1. The hormone concentrations (DHT and R1881) must be added for each experiment. Unclear is whether supraphysiological levels were used. Page 4. Define androgen-starved. Is charcoal treatment meant or FBS-free media?

Response: Throughout our experiments we used 10 nM DHT for androgen stimulation, unless otherwise indicated. We have clarified this in the revised text. This is above physiological levels and is generally used to assess effects of maximal stimulation, although as DHT is rapidly metabolized the effective concentration at longer time points is lower. We do show in Figure 2 that the suppressive effects on MYC occur at 0.1 – 1 nM, within the physiological range. We now also clarify on page 4 that cells are androgen starved by culturing in medium with FBS that has been charcoal dextran treated to remove steroids “(...datasets comparing androgen-starved cells (cultured in medium with FBS that is charcoal dextran stripped to deplete steroids, CDS medium) versus DHT-stimulated cells...”.

2. Missing are the transcriptome and ChIP-seq analysis of the cell lines under androgen-starved condition. Are more AR binding sites occupied in AR high expressing cells? Please show the basal level data without addition of androgens.

Response: We have now added the analysis of AR ChIP-Seq under vehicle condition in LNCaP (AR-low) and VCaP (AR-high) cells to Supplementary FigureS1C-E. Indeed, the AR high expressing cells have more AR binding under androgen-starved (35904 AR peaks in VCaP and 536 in LNCaP) as well as DHT-stimulated conditions. The raw transcriptome data under androgen-starved and DHT-stimulated conditions have been deposited, and what we describe in the manuscript is the change in expression in response to DHT.

3. Authors used samples of CRPC patients. Unclear is whether the patients underwent ADT prior the analysis. Authors should make a statement for this issue. Authors imply that since in CRPC MYC is overexpressed, MYC to be a key factor in CRPC.

Response: Our clinical analyses were based on CRPC samples from two reports. The gene expression analyses were based on 101 castration-resistant prostate cancer metastases (PMID: 30340047). The

epigenetics study was based on 15 PDX tumors derived from patients with mCRPC (PMID: 32690948). Therefore, all patients underwent ADT prior to the tissue being obtained. This has now been clarified in the text on page 13 “For this analysis we compared gene expression in primary untreated PCa versus samples from men taken who had progressed after androgen deprivation therapy (castration –resistant prostate cancer, CRPC)”, and figure legends.

4. Authors reveal that loss of MYC accounts for 8% of DHT repressed genes. In this context authors should tune down their too generalized statements about DHT mediated-repression of gene expression is mediated by MYC. Also the statement on page 9 should be tuned down: “MYC may have a large impact on the AR transcriptome”.

Response: We agree that accounting for just 8% of DHT-repressed genes should not qualify as a large impact, and we have modified the previous text in this section to state “...these results support MYC downregulation as a mechanism that contributes to DHT-mediated transcriptional repression”. Notably, in addition to effects related to DHT-repressed genes, we further show that sites with overlapping AR and MYC sites have increased H3K27ac versus AR unique sites, and are similarly more frequently linked to genes that are altered in response to DHT. Therefore, its effects on the AR transcriptome go beyond the 8% of DHT-repressed genes. This is further clarified in the text related to figure 3.

5. These gene set of 8 % should be analyzed in more detail to identify pathways.

Response: As suggested by the reviewer, we performed KEGG pathway and GO enrichment analyses using the common genes (the 8% of DHT-repressed genes that overlapped MYC stimulated genes) in Figure 2M. While no KEGG pathway was enriched, we found two Gene Ontology Biological Processes terms (“pseudouridine synthesis” and “mRNA modification”) specifically enriched in these common genes (Supplementary Figure S2M). This enrichment result suggests the genes downregulated by DHT as well as upregulated by MYC play a role in RNA modification. We thank the reviewer for the suggestion, and this is an interesting observation to pursue in future work.

6. Please show data whether change of BRD4 recruitment correlate to AR repressed genes and analyze pathways of this subset of genes in IPA or GSEA.

Response: We also thank the reviewer for this suggestion. We have now added these data to Supplementary Figure SIG-I. The analyses showed that BRD4 recruitment was decreased by DHT treatment at AR repressed genes but increased by DHT treatment at AR activated genes (Figure SIG). To determine whether the DHT-mediated decrease in BRD4 binding was affecting a functionally related set of genes, we identified the subset of DHT-repressed genes that had the greatest decrease in BRD4 binding (Figure S1H). This resulted in 91 genes with 30% reduction of BRD4 binding from 567 DHT-repressed genes. We then performed GO BP and canonical pathway enrichment analyses using these genes and gene sets from MSigDB/GSEA (Figure S1I). Interestingly, gene ontology biological process and canonical pathway enrichment analyses suggested involvement in regulation of apoptosis, which may be pursued in future studies.

7. Authors focus on one pioneering factor FOXA1. Authors should analyze their ChIP-seq data genome-wide also for sequence binding motifs of the additional two known pioneering factors HOXB13 and GATA2.

Response: As suggested, we checked the enrichment of HOXB13 and GATA2 motifs in MYC and AR peaks of Figure 3B. Notably, both HOXB13 and GATA2 motifs are only enriched in AR unique peaks, but not in

AR/MYC common peaks or MYC unique peaks. This interesting result has been included in Supplementary Figure S3A, and further supports novel features of the AR/MYC shared sites.

8. Please show the knockdown of AR at protein level.

Response: Unfortunately we do not have the AR protein level data as the results shown in Figure S3G and S3H are based on an analysis of a previously reported data set from another group where AR siRNA was used to decrease AR (Misawa A, Takayama K, Urano T, Inoue S. Androgen-induced Long Noncoding RNA (lncRNA) SOCS2-AS1 Promotes Cell Growth and Inhibits Apoptosis in Prostate Cancer Cells. J Biol Chem 2016 Aug 19;291(34):17861-80. PMID: 27342777; Takayama KI, Fujimura T, Suzuki Y, Inoue S. Identification of long non-coding RNAs in advanced prostate cancer associated with androgen receptor splicing factors. Commun Biol 2020 Jul 23;3(1):393. PMID: 32704143) (GSE82223). The siAR was from ThermoFisher/Ambion (Catalog s1538) that targets AR exon-2. The target sequence is: ttgcccattgactattac, and its specificity and uniqueness were confirmed by BLAST against human genomic + transcript databases. Complementary to its knock-down effects on AR message in the RNA-seq (GSE82223), this siAR was also shown to specifically attenuate AR mRNA in RT-PCR analyses (PMID: 27342777) and AR proteins in Western blotting (PMID: 30649323), all published by the same lab. We now further clarify these points in the text and figure legend. Moreover, as we do not show effects on AR protein, we have modified the text to state that the repression was prevented by treatment with AR siRNA (rather than stating AR depletion) : “Analysis of a previously reported data set (GSE82223) confirmed that the MYC-activated gene set was repressed by DHT in VCaP cells, and this repression was prevented by treatment with AR siRNA (Figures S3G and S3H)”.

Minor points

Authors used knockdown and not knockout. Therefore, please replace “loss of MYC” by “reduced MYC”

Response: We have replaced loss with reduced.

Page 4: “Androgen stimulation” should be replaced. E.g. androgen treatment or androgen-mediated stimulation. Androgen itself cannot be stimulated.

Response: This has been changed.

Page 17: replace “DHT repression of MYC” by DHT-mediated repression of MYC activity.

Response: This has been changed.

Authors state that MYC inhibits AR very generally driven by coactivator redistribution, although only one specific (BRD4) is meant. Please tune down some sentences generalizing coactivators.

Responses: We presume that the effects we see are not exclusively due to BRD4 redistribution, but agree that our generalization should be tuned down since our data only directly examined BRD4. This has now been done in the revised text.

Many sentences should be shortened to avoid misinterpretation.

Response: We have done further editing to shorten some sentences and avoid confusion.

Reviewer #2, expert in super-enhancers (Remarks to the Author):

The manuscript by Guo H et al. elucidated the intricate interactions between AR and MYC in prostate cancer. They both compete for coactivators and function cooperatively to maintain a stable expression of genes regulating multiple cellular functions. Their findings showed that increased MYC in response to androgen deprivation contributes to the development of CRPC, while decreased MYC may contribute to tumor regression in response to supraphysiological androgen therapy in men with CRPC. The authors performed a wealth of bioinformatic analyses to support their findings, which are biologically interesting and clinically relevant. However, substantial effort should be made to improve the clarity of the manuscript by providing more technical details about various analyses and proofreading by a native speaker.

Major comments:

1. A lot of bioinformatics analyses were performed throughout the manuscript, but substantial clarifications should be made. The description of Methods should be more detailed. For example, the trimming steps in the sequencing data analysis, the software/package version information, as well as the detailed specifications of parameters/arguments should be provided. The statistic tests used should be provided and properly phrased to support the conclusions, e.g., Figure 1C, 1D, Figure 4B-E, G, H, J, K, Figure 5A-B, I, Figure 7C, D, and G.

Response: We thank reviewer for pointing out this issue, which is important for the scientific accuracy of our manuscript. In the updated manuscript, we added the description of sequencing data trimming as well as the detailed parameter setting of analysis tools for RNA-seq, ChIP-seq and HiChIP to the Methods. We also conducted statistic tests for the RT-qPCR, ChIP-qPCR and cell proliferation results throughout all the figures, and added the description of statistic methods and cut-off to the figure legends. Per journal policy, we also provided the source data for the above analyses.

2. Specifically in Figure 1:

1) Figure 1C-D: what is the method or tool for identifying Androgen activated and repressed genes using the VCaP RNA-seq data?

Response: Originally, we identified androgen activated and repressed genes in Figure 1C-D using Wilcoxon rank sum test of FPKM values. In order to identify differentially expressed genes (DEGs) by FDR (or adjusted P-value) as suggested by the reviewer, we re-ran the DEG analysis using DESeq2 package and RNA-seq read count matrix. We have updated Figure 1C-D with DESeq2 results.

How consistent are the differential/dysregulated genes identified by microarray and RNA-seq data in LN, VC and VA?

Response: We now compared the DEGs from microarray and RNA-seq data in Supplementary Figure S1A. Comparison of the shared genes also showed that the number of DHT-repressed genes was increased in the VCaP and AR-overexpressing LNCaP cells. The results also show that the upregulated genes by DHT are more consistent in microarray and RNA-seq data in all three cell lines than the downregulated genes, likely due to greater changes in the upregulated genes. Although not requested, we also examined the RNA-seq data for overlap in genes that were upregulated or downregulated in each cell line, which also showed greater effects in the VCaP and LNCaP-AR cells (Figure S1B).

Why different cutoffs on fold changes were used (1.5 and 2)?

Response: During the bioinformatics analysis, we found RNA-seq data are generally more sensitive than microarray data for the detection of DEGs at the fold change level. While we use a less stringent fold

change cutoff (1.5) for microarray data, we still got many fewer DEGs in microarray data than RNA-seq data. If we use the same fold change cutoff for both microarray data and RNA-seq data, the numbers of DEGs are either too small for microarray data by fold change > 2 or too large for RNA-seq data by fold change > 1.5. Therefore, to get a reasonable number of DEGs in both microarray data and RNA-seq data, we decided to use different cutoffs on fold changes. Notably, the comparisons in Figure 1C are based on microarray data in each cell line or RNA-seq data in each cell line, and not microarray versus RNA-seq with distinct thresholds.

Also, the common cutoffs on the statistical significance for selecting differentially expressed genes should be $FDR < 0.05$, instead of $P \text{ value} < 0.05$.

Response: As suggested, and as noted above, we now use FDR with the adjusted P value < 0.05 as the statistical significance for DEG selection. To do that, we ran DEG analysis pipelines using limma package for microarray data and DESeq2 package for RNA-seq data.

2) Figure 1E: AR binding peaks were integrated with differential expression for the analysis, but there was no data showing the binding patterns of AR. Were the binding peaks identified using ChIP-seq data at 2hr, 10hr and 24hr, respectively?

Response: Thank you for bringing this up as we agree it is not clear in the text. AR genomic occupancy in the current study was based on the reported VCaP AR ChIP-seq datasets from the Chinnaiyan lab (GSE55062, Asangani et al., 2014, Nature). In these studies, VCaP cells were grown in medium with steroid-depleted serum for 48 hrs and then treated with 10 nM DHT for 12 hr. Notably, although AR chromatin binding is a dynamic process and peak intensities may fluctuate, previous studies have shown that the overall binding profile is relatively persistent over about 24 hr. We have now added this information to the text and figure legend. Our finding of a weak association of AR with androgen-repressed genes is also consistent with our previous report, which was based on an AR ChIP-Seq in VCaP at a different time point (DHT, 4hrs; GSE32345; PMID: 27760327).

More information should be provided to illustrate/compare the genome-wide distributions of AR binding sites (promoters, gene body, intergenic, etc.) in LN, VC and LA cell lines.

Response: We have now included the genome-wide distributions of AR peaks in LNCaP and VCaP cells in Supplementary Figure S1D. However, there is no published AR ChIP-Seq data in LNCaP-AR cells.

3) Figure 1F: please explain what it means by “genes classified based on the distances to AR and H3K27ac (GSE96652) peaks”.

Response: We apologize for the lack of clarity. We have now modified the Figure 1F legend as: “Boxplots show expression fold changes of genes, which were classified into AR only, H3K27ac only, H3K27ac/AR common or None groups based on H3K27ac or/and AR occupancy at $\pm 10\text{Kb}$ regions of gene TSSs”.

Also, the author showed the expression of DHT-repressed genes that are associated with H3K27ac sites that do not overlap AR sites (H3K27ac_only) decreased most significantly in response to DHT. Are there any DHT-repressed genes that are not associated with H3K27ac and AR sites?

Response: There are some DHT-repressed genes that are not associated with H3K27ac or AR binding, although they may have enhancers located more distally. We have now added the genes without AR or H3K27ac binding at their $\pm 10\text{Kb}$ regions to Figure 1F. As shown, DHT-repressed genes that are not

associated with H3K27ac and AR sites decreased least among all the four groups.

3. In Figure 6B, the author showed that “The PCAT1 SE has three smaller sub-SEs, and H3K27ac at each site and overall was markedly decreased upon DHT stimulation”. A statistic test should be provided to support the “markedly” conclusion since the figure only showed the log₂ fold changes and average signals of super-enhancers.

Response: To address this concern, we performed statistical tests of the three PCAT1 sub-SEs between vehicle condition and 2hr-DHT treatment. Each PCAT1 SE was divided into 100 bins, and the average signal of each bin was extracted from bigWig files for box plots and paired t-tests. This analysis showed a very significant decrease of H3K27ac signal in all three sub-SEs ($p < 2.2e-16$, $p < 2.2e-16$, and $p = 2.7e-12$). The results have been included as Supplementary Figure S6B.

4. The authors should carefully check all the terms and abbreviations, and keep them consistent throughout the manuscript, such as “androgen-repressed genes”, “DHT-repressed genes” and “DHT-downregulated genes”, “LNCaP-AR” and “LA”, “VCaP” and “VC”, “ChIP-Seq” and “ChIP-seq”. Some ambiguous terms may lead to misunderstanding, such as “AR activity” in “AR activity at MYC independent genes”, “regulatory interaction” in “Locus-wide repression of the 8q24 TAD by androgen correlates with the decline in regulatory interaction”.

Response: We apologize for inconsistencies in the text, and have done further editing in efforts to address them in the revision. In particular, we now primarily use DHT-repressed to describe results where we treat with DHT.

Minor comments:

1. Figure 2A-B: how were the 1875 human TFs selected? A proper citation of the literature or database is needed.

Response: We selected the human TFs based on a review by Vaquerizas JM et al, 2009, Nat Rev Genet (PMID: 19274049). We have now added the citation of this paper.

2. In Figure 5G, there is no difference in the size of the dots denoting KEGG pathways with gene ratios of 0.05 and 0.10.

Response: We thank the reviewer for pointing this out. We re-generated the KEGG enrichment plot and updated Figure 5G.

3. Typos, e.g.:

1) AR dives terminal differentiation. “dives” -> “drives”?

2) In the Method section “TCGA PCa and Quigley CRPC RNA-Seq data analysis”, “TMP values” should be “TPM values”.

3) “MYC up-regulated” should be “MYC upregulated” to be consistent with “DHT_downregulated”.

Response: We thank the reviewer for picking these up and have revised these errors accordingly.

Reviewer #3, expert in ChIP-seq and RNA-seq analysis (Remarks to the Author):

In this manuscript, the authors showed that DHT treatment repressed MYC expression and androgen deprivation increased MYC expression. They concluded that increased MYC in response to androgen

deprivation contributes to castration-resistant prostate cancer, while decreased MYC may contribute to responses to supraphysiological androgen therapy. The novel finding such as super-enhancer relevant results in Figure 6 and Figure 7 is interesting. The authors could emphasize this major finding and add more mechanism results in the main Figures to enhance the novelty in this paper. However, most the phenomenon in the manuscript has been reported. For example, Lam et al (Eur Urol. 2020 Feb;77(2):144-155.) reported that supraphysiological testosterone treatment impaired cell cycle via robust downregulation of Myc-E2F pathway in ENZR PDXs. Monga et al (Sci Rep. 2020 Apr 20;10(1):6649) showed that c-MYC was upregulated in enzalutamide-resistant prostate cancer. Therefore, the authors should make effort to gain more novel mechanistic insight in the main Figures to specify the new findings. Overall, although the phenomenon and clinical significance reported in the manuscript are not very new, the equilibration of AR and MYC during anti-AR therapy and supraphysiological androgen therapy is interesting. The author could specify the finding and enhance the logic to present data in the main Figures.

Response: We thank the reviewer for these comments. We agree that previous studies have shown downregulation of MYC by androgen, and that the novelty of our study is in the identification of mechanisms. We have done further editing to highlights these novel aspects.

1. In Figure 1C, the authors concluded that the number of androgen-activated/repressed genes in LNCaP cells is lower than in VCaP cells and AR-overexpressing LNCaP cells. However, there are some concerns for these data. Firstly, the microarray data and RNA-seq data of LNCaP, VCaP, and AR-overexpressing LNCaP cells were not acquired at the same conditions. The androgen concentration/treatment time, cell culture condition and microarray/RNA-seq platforms may influence the results which could affect the conclusion. For example, this manuscript shows that the number of androgen-LNCaP-activated/repressed genes in LNCaP cells is about 200, but Wang et al (Cell. 2009 Jul 23;138(2):245-56) reported that the number of DHT upregulated genes in LNCaP cells is about 400. If the authors chose this microarray data, the conclusion might be different. Therefore, the authors should be cautious with their conclusion.

Response: We are in full agreement with the reviewer that experimental conditions and methods can have profound effects on AR protein levels and chromatin state, and thus the outcomes of androgen-mediated gene profiling. Indeed, new data added to Supplementary Figure S1 clearly show that overlap between genes identified by microarray and RNA-seq in each cell line only partially overlap, and that genes identified by RNA-seq across the cell lines similarly only partially overlap (Figure S1A and S1B). Therefore, it is clear that the number of genes we report as being DHT-stimulated or DHT-repressed is not precise. Nonetheless, the differences we see in both the microarray and RNA-seq data, or in examining genes that overlap in the microarray and RNA-seq data, do show greater effects in the cells expressing higher levels of AR. Moreover, as the focus of this study is on MYC, we do show by direct comparison under identical conditions that DHT repressed MYC expression in the VCaP cells and LNCaP cells overexpressing AR, but not in parental LNCaP cells (Figure 2).

2. Figure 1F and Figure 1G are confusing. In Figure 1F, the P value looks very significant in the middle panel, but the fold change is minimal.

Response: Since we used all genes in the middle panel of Figure 1F, the numbers of genes in each group are very large. During the statistics test, large sample numbers could generate very significant P-values even if the fold change is small. For the reviewer's reference, the average fold changes in the middle panel are -0.11 and 0.016 for H3K27ac only and H3K27ac/AR common groups respectively, and the gene numbers are 25829 and 1876 for H3K27ac only and H3K27ac/AR common groups respectively.

In Figure 1G, what is the value "Change = 0.0239". What does it mean? It is the fold change or p = value? I

also didn't get the main point about Figure 1G. The authors should make it clearer and easier to understand.

Response: The value "Change = 0.0239" indicates the relative signal change at the center of curves. It is very confusing as pointed by the reviewer. We now replaced the "Change" with the P-values calculated by Student's t-tests using the ChIP-seq signal at the center of each peak. The comparisons of ChIP-seq signal at the peak centers are also illustrated by box plots in Supplementary Figure S1F.

3. In Figure 2D- F, could the authors comment on why DHT regulated MYC expression in VCaP cells but not in LNCaP cells? It is because AR level is very low in LNCaP cells? If so, AR level should be negatively correlated to MYC expression. Is this true in the patient's database such as TCGA?

Response: Our interpretation is indeed that DHT represses MYC in VCaP cells, but not (or to a much lesser extent) in LNCaP cells due to high AR in VCaP cells. This interpretation is further supported by the repression of MYC in the LNCaP cells overexpressing AR (Figure 2F and 2G). Assessing correlations between MYC and AR in clinical samples is challenging as the AR gene is frequently amplified in CRPC and AR mRNA or protein do not provide accurate metrics of AR activity (as androgen-liganded AR can feedback and suppress AR activity). Similarly, the MYC gene is also frequently amplified in CRPC, and many other factors can regulate MYC expression. Nonetheless, when we exclude cases with AR and MYC gene amplification, we do find that MYC is increased in CRPC (where AR activity is decreased) (Figure 5J). As suggested by the reviewer, we also examined AR versus MYC in the TCGA dataset of primary PCa. Notably, while there is a positive correlation between AR and MYC mRNA (left panel), there is a negative correlation when comparing MYC with AR regulated genes (middle and right panels). We have not included these data in the manuscript as the negative correlations are modest, but they are consistent with MYC repression being mediated by the activated AR.

4. The author showed that all MYC binding sites were decreased when treating with DHT (Figure 2J). However, in Figure 2M, why most of MYC up-regulated genes are not overlapped with DHT downregulated genes?

Response: This is a good point that we should have noted and addressed in the original text. Indeed, given the robust DHT-mediated decrease in MYC, one might expect to see more substantial overlap between MYC upregulated and DHT-repressed genes. One important difference is that for the DHT treatment we examined genes that were repressed within 24 hours. In contrast, for the MYC siRNA treatment we examined effects on gene expression after 3 days, suggesting that many of the genes decreased by MYC siRNA may not be suppressed by acute decreases in MYC, but decline more slowly after MYC downregulation. In the revised manuscript we now note this less than expected overlap, and suggest timing as a mechanism; "Conversely, this group of overlapping genes reflected ~20% of the genes that were decreased by MYC siRNA. This latter overlap with genes repressed by MYC siRNA may be less than expected given the marked DHT-mediated

decrease in MYC. However, this ~20% may reflect genes that are most acutely (within 24 hours) altered by decreased MYC (and not directly or indirectly stimulated by AR), versus those that are decreased by the siRNA-mediated decrease in MYC over 2 days". Notably, we clearly can't rule out other mechanisms including indirect effects of decreased MYC over 2 days, or compensatory effects of increased AR activity, which may be further explored in future studies.

5. The statistical significance in the bar graphs such as Figure 1A, Figure 2D and others should be labeled.

Response: In the revised manuscript, we include statistics for all bar graphs and also include all source data.

Reviewer #4, expert in MYC signalling (Remarks to the Author):

Androgen Receptor and MYC Equilibration Centralizes on Developmental Super-Enhancer By Guo et al. In their manuscript, Guo and colleagues provide a detailed analysis of the epigenetic mechanisms governing the androgen receptor-MYC signaling axis in prostate cancer. To this end they perform integrated analyses of newly generated ChIP-seq and RNA-seq data of cell line based prostate cancer models and employ publicly available datasets to support their observations. They thereby provide insights into the PCAT1 super-enhancer driven expression of MYC in prostate cancer, which is abrogated by AR activation leading to MYC repression and BRD4 redistribution. It is suggested that these processes thus indirectly promote the lower expression of genes suppressed by DHT treatment, which are a primary focus of the investigation. Overall, the in vitro experiments are well conducted, include relevant controls and appear technically sound. Comparable effects are observed in a cell line xenograft.

However, the novelty of the results is limited by the fact that several previous papers have investigated the connection between AR and MYC signaling in prostate cancer. Without being an expert in the prostate cancer field, it has been already reported in detail how MYC overexpression reprograms AR chromatin occupancy (Barfeld et al. EBioMedicine 2017). In addition, there is apparently some controversy in the field with respect to the role of MYC as there is no effect on AR as shown in a mouse model (Kim et al. Oncogene 2012) and in cell lines models AR knockdown has been shown to reduce the expression of MYC (Gao et al. Plos One 2013). Taking into account the body of published data the excitement for the presented data is limited. Also, while the data presented provide insight into the regulation of gene expression following DHT treatment and AR-MYC interplay several points remain to be further clarified, especially regarding the mechanism by which AR leads to MYC repression without DNA binding or dimerization capacity.

Response: The reviewer is correct that effects of MYC overexpression on AR have been described previously (although with some conflicting data), but the focus of this study is on AR regulation of MYC. Also as noted by the reviewer, in some contexts the loss of AR may decrease MYC. This is quite interesting and may reflect a role for AR in opening what appears to be a prostate cancer specific (and possibly developmentally regulated) MYC superenhancer that overlaps PCAT1. In contrast, studies from a number of other groups (including our own previous data) have found that androgen treatment represses MYC expression. However, the basis for this repression has not been determined, and this is the major focus of the current study.

Major points:

1. It seems puzzling in Fig. 7F/G that AR is able to suppress MYC expression even in the absence of DNA binding or dimerization capacity (M1-3), but fails to do so if nuclear localization is mitigated (M4). Several additional experiments would help to elucidate the underlying biology:

- Transcriptomic data of LNCap wt, M1-4 +/- DHT would be very helpful to assess the overall effect of the mutants on AR signaling and e.g. loss of induction of AR stimulated genes in M1-3 despite effects on MYC.

- Can it be shown that DNA binding/dimerization are indeed defective for M1-3 in the context presented?
- Enzalutamide blocks binding of androgens to AR, binding of AR to DNA and nuclear translocation of AR. It would be helpful to see whether AR translocation is inhibited by enzalutamide in Fig. S2B/5A e.g. by IF. Do the same results with alternative AR inhibitors that do not prevent nuclear translocation (e.g. bicalutamide)?

Response: We thank the reviewer for these excellent suggestions. We have now examined the effects of the mutant ARs, versus the wild-type (WT) AR on expression of the strongly AR-stimulated PSA/KLK3 and TMPRSS2 genes (Supplementary Figure S10B). As expected, the M1-3 mutants do not enhance AR activity, and instead appear to markedly impair the activity of the endogenous AR. Notably, the M4 mutant, which is defective in nuclear translocation, does not repress TMPRSS2 and is less repressive on PSA/KLK3. The basis for differential effects of overexpression of WT and M4 AR on PSA/KLK3 versus TMPRSS2 are not clear, but likely reflect distinct cofactors for the PSA/KLK3 superenhancer. In any case, these results are consistent with previous studies showing defects in DNA binding or nuclear localization for these mutants. They are also consistent with the effects of these mutants on MYC, and support the conclusion that nuclear AR can suppress MYC expression independent of DNA binding.

Enzalutamide can impair DHT-stimulated AR nuclear translocation. However, most studies show that enzalutamide treatment (in the absence of DHT) does not block AR nuclear localization, but it does prevent coactivator binding and recruitment of AR to specific binding sites on chromatin. Therefore, we hypothesize that that enzalutamide does not repress MYC primarily because it does not support coactivator recruitment. As suggested by the reviewer, we have now addressed the effects of an alternative AR inhibitor, bicalutamide. In contrast to enzalutamide, the bicalutamide-liganded AR can bind to androgen-responsive elements on chromatin, but similarly to the enzalutamide liganded AR it does not effectively recruit coactivators. Therefore, we tested the effects of bicalutamide on MYC expression. As expected, bicalutamide did not stimulate PSA, and notably did not repress MYC (Supplementary Figure S5A and S5B). These findings support the conclusion that MYC repression is dependent upon coactivator competition.

2. It is suggested that MYC and AR compete for co-activators like BRD4 in the context of DHT treatment.

- Can overexpression of co-activator BRD4 mitigate the effects on target genes expression during DHT treatment?
- Fig. S10A suggests that AR not bound to DNA may still sequester co-activators, which seems in line with results from DNA binding defective variants (Fig. 7G). Can binding of non-DNA-bound AR to e.g. BRD4 be shown (e.g. co-IP or PLA +/- DHT)?
- Fig. 7C: Neither JQ1 nor DHT appear to strongly reduce BRD4 at the PCAT1 SE site and also lack a formal test of significance compared to vehicle. This effect appears much smaller than the effect observed on MYC expression. Please explain?

Response: It should be emphasized that while the current study focused on BRD4, we hypothesize that DHT-mediated transcriptional repression of MYC (and additional genes) in cells expressing high levels of AR is due to redistribution of multiple cofactors. Indeed, despite the potent effects of JQ1 on blocking BRD4 binding to chromatin and repression of MYC expression, there is further repression with the addition of DHT (Fig.7D). As noted below, in new data we also show an interaction with MED1 that is not dependent on AR DNA binding. Therefore, given the myriad of transcriptional coactivators recruited by AR, we have not carried out studies to assess their relative contributions.

Previous studies have also shown that nuclear AR is localized in foci containing transcriptional coactivators (such as CBP) prior to its chromatin binding, and that AR mutants defective in DNA binding are still localized in these foci (Marcelli M et al., Quantifying effects of ligands on androgen receptor nuclear

translocation, intranuclear dynamics, and solubility. J Cell Biochem. 2006 Jul 1;98(4):770-88. PMID: 16440331 (Black BE et al., Transient, ligand-dependent arrest of the androgen receptor in subnuclear foci alters phosphorylation and coactivator interactions. Mol Endocrinol. 2004 Apr;18(4):834-50. PMID: 14684849). This would be consistent with our finding that AR mutants defective in DNA binding can still repress MYC expression. As suggested by the reviewer, we have now directly tested by coimmunoprecipitation whether the well-characterized M2 mutant (AR-C619Y) could still interact with BRD4. As shown in Supplementary Figure S10C, immunoprecipitation of the Flag tagged WT and M2 mutant AR from stably transfected LNCaP cells (same cell lines used to show MYC repression) brought down comparable levels of BRD4. We similarly observed interactions with another critical co-activator, MED1. We also assessed for CBP and p300, but could not see clear bands for the WT or mutant, presumably due to weaker transient interactions (not shown). This result for BRD4 is consistent with a previous report indicating that BRD4 interacts with the AR N-terminal domain (Asangani IA, Therapeutic targeting of BET bromodomain proteins in castration-resistant prostate cancer. Nature. 2014 Jun 12;510(7504):278-82; PMID: 24759320).

In Figure 7C, we agree that the decrease in BRD4 binding in response to JQ1 and DHT is less than one might expect, and this has been pointed out in the manuscript. With respect to DHT, we have now tested for significance and the DHT-mediated reduction is significant. Moreover, it is not inconsistent with the decrease seen by BRD4 ChIP-seq, where one can still see a reduced peak at this site after DHT treatment (Supplementary Figure S9C). The greater loss of MYC binding at this site is not surprising, as the basis for this loss is the dramatic decrease in MYC protein, versus the decrease in BRD4 binding due to redistribution. Finally, the observation that substantial BRD4 binding persists after DHT-treatment is consistent with a role for redistribution of additional coactivators, and/or with the possibility that this MYC enhancer is very sensitive to even modest decreases in BRD4 binding (particularly as this decrease is occurring at several sites in this superenhancer). With respect to effects of JQ1, it is generally very effective at displacing BRD4 from chromatin, so we do not have a compelling explanation for why it did not cause more robust loss of BRD4. Notably, it did have a rapid and marked effect on suppressing MYC mRNA in the same VCaP cells, so the drug was clearly active (Figure 7D). Our current hypothesis is that BRD4 binding at this particular site is relatively stable with a slow off rate, and that the rapid and robust effects of JQ1 on MYC expression reflect the effects of modest loss at this site combined with losses at additional sites. We now point out these modest effects on BRD4 binding in the revised manuscript, and suggest that the combined effect of modest losses at several sites may be the basis for the marked decrease in MYC in response to JQ1.

3. What is the translational relevance of these findings? This a topic that has not been touched too much although some of the findings might be of therapeutic relevance such as the increased binding of BRD4 at AR sites upon DHT treatment. This would be a major point to increase the impact of the study and to enhance its novelty.

Response: The importance of MYC as an oncogene is well-established, but it has been challenging to develop therapeutic approaches that can target MYC. Androgen deprivation is the standard therapy for metastatic PCa, but virtually all patients eventually progress. This study is highly translationally relevant as it directly links androgen deprivation to increased MYC, and implicates this increase as one factor contributing to the ultimate failure of androgen deprivation. Therapeutically, a prediction would be that short-term treatment with an agent targeting MYC (when available) during the initiation of androgen deprivation therapy may in particular enhance efficacy. Conversely, there is growing interest in the use of supraphysiological testosterone (SPT) therapy for men who progress to castration-resistant prostate cancer (CRPC). This study has further translational relevance as it shows that SPT is a therapeutic approach that can suppress the expression of MYC and multiple other genes through cofactor redistribution. We anticipate this suppression creates vulnerabilities that may be exploited by combination therapies, and this is a

direction we are currently pursuing. We have further emphasized these points in the revised text.

Minor points:

1. Fig. 4A-E: Based on LA and LAM MYC expression decreases AR-only genes KLK3 and TMPRSS2. Investigation and side-by-side comparison with parental LNCap or ideally LNCap-MYCoE would be helpful to further judge the MYC AR interplay.

Response: As suggested by the reviewer, we analyzed the AR-activated gene set in transcriptomic data from LNCaP cells overexpressing MYC (LNCaP_MYC_OE) versus LNCaP cells, using a previously reported data set from the Mills group (GSE73917). Consistent with the conclusion from their study (PMID: 28412251), and with our findings in LNCaP cells overexpressing AR, this analysis showed a very clear decrease of AR signature gene expression in the LNCaP_MYC_OE cells (Supplementary Figure S4D). These results indicate that increased MYC can suppress AR activity both in cells expressing basal and high levels of AR.

2. An overview/table of the data sets used indicating cell line/sample, treatment (compound, concentration, time), investigated marker, technology and source would be helpful.

Response: We appreciate this excellent suggestion and a Table with this information has been added to the revision as Supplementary table 2.

3. The impact on cell proliferation of DHT/Enz has been shown for some of the cell lines used. To get a clearer picture about the impacts on global cellular phenotype viability assays assessing the impact of DHT concentrations and androgen depletion (CDS medium) on cell growth in VCAP, LNCaP, LA, LAM and LAM1-4 would be helpful.

Response: Previous studies from many groups have shown that PCa cells including LNCaP and VCaP have biphasic responses to DHT. Growth is suppressed (with cells arrested in G0/G1) in androgen depleted medium or by treatment with Enz, and restoration of androgen at low levels can stimulate cell cycle progression. In contrast, this stimulation is lost when cells are treated with higher concentrations of androgens. The optimum stimulatory concentration of androgen varies with the cell line and exact culture conditions. There are likely several factors contributing to this biphasic response, and our focus in this study is on the contribution of decreased MYC. In Supplementary Figure S10E we show that the growth of parental LNCaP cells in CDS is stimulated by DHT (10 nM), and this is associated with an increase in MYC. In contrast, the growth of LNCaP cells in medium with androgen (FBS) is suppressed by addition of 10 nM DHT, with a modest decrease in MYC. In Supplementary Figure S10F we then show that the overexpression of AR in LA cells enhances the growth suppressive effect of DHT, and this is associated with a marked decrease in MYC. Finally, the overexpression of MYC in the LA cells (LAM cells) prevents the decrease in MYC in response to DHT, and partially prevents the decrease in cell proliferation. We believe these data provide picture of the effects of AR and MYC on response to androgen, and support the conclusion that decreased MYC contributes to the growth suppressive effects of high androgen levels.

4. If loss of MYC expression contributes to 7-8% of DHT suppressed genes, by what is the rest suppressed by?

Response: We presume the redistribution of BRD4 and other transcriptional cofactors that decrease MYC is also directly decreasing expression of many of the DHT-repressed genes. Moreover, in addition to MYC, some of these decreased genes are transcription factors, so their decrease would be a further indirect mechanism contributing to DHT-mediated transcriptional repression. Finally, AR may be acting directly as a transcriptional repressor on a subset of genes.

5. Both amplification of AR and of MYC occur in CRPC. Are these independent events or do they co-occur?

Response: Biologically these appear to be independent events. MYC amplification occurs in a subset of primary PCa, and is associated with higher grade tumors. AR amplification is very rare in primary PCa, but is very common in tumors that recur after androgen deprivation therapy (castration-resistant prostate cancer; CRPC). The basis for this increase in AR amplification is selective pressure to maintain AR activity in the setting of decreased androgen levels. Due to the high frequency of AR amplification and of MYC amplification, there is substantial overlap. To assess this further, we analyzed the co-occurrence of AR and MYC amplification in the CRPC samples from a recent study (Quigley, D.A. et al. Genomic Hallmarks and Structural Variation in Metastatic Prostate Cancer. Cell 174, 758-769 e759, 2018). As expected there was substantial overlap between AR and MYC amplification, but it is not statistically significant as determined by hypergeometric distribution.

Reviewers' Comments:

Reviewer #1:

Remarks to the Author:

Authors have addressed the critical points in a full satisfactory manner.

Reviewer #2:

Remarks to the Author:

My previous concerns have been satisfactorily addressed, and I have no further comments.

Reviewer #3:

Remarks to the Author:

The authors performed a series of experiments and further analyses of the data and have adequately addressed all my concerns. The new data support the conclusion. The revised manuscript has been improved significantly and is acceptable for publication in Nature Communications.

Reviewer #4:

None

Response to reviewer comments

We thank the reviewers for their previous helpful suggestions. There are no further comments to address.